# Short tandem repeats delineate gene bodies across eukaryotes

William B. Reinar [1,2] ✉, Anders K. Krabberød [1,2], Vilde O. Lalun[1,2], Melinka A. Butenko[1,2] & Kjetill S. Jakobsen [1] ✉

Short tandem repeats (STRs) have emerged as important and hypermutable sites where genetic variation correlates with gene expression in plant and animal systems. Recently, it has been shown that a broad range of transcription factors (TFs) are affected by STRs near or in the DNA target binding site. Despite this, the distribution of STR motif repetitiveness in eukaryote genomes is still largely unknown. Here, we identify monomer and dimer STR motif repetitiveness in 5.1 billion 10-bp windows upstream of translation starts and downstream of translation stops in 25 million genes spanning 1270 species across the eukaryotic Tree of Life. We report that all surveyed genomes have gene-proximal shifts in motif repetitiveness. Within genomes, variation in gene-proximal repetitiveness landscapes correlated to the function of genes; genes with housekeeping functions were depleted in upstream and downstream repetitiveness. Furthermore, the repetitiveness landscapes correlated with TF binding sites, indicating that gene function has evolved in conjunction with *cis*-regulatory STRs and TFs that recognize repetitive sites. These results suggest that the hypermutability inherent to STRs is canalized along the genome sequence and contributes to regulatory and eco-evolutionary dynamics in all eukaryotes.

Tandemly repeated DNA motifs are intrinsic to eukaryotic DNA sequences, both as the primary component of long satellite DNA sequences and as shorter repetitive stretches among regions of unique sequence[1–8]. Short tandem repeats (STRs) have been detected in all examined eukaryotic genomes, and their genomic proportions are correlated with ecology and genome size in certain taxa[7,8]. Monomer STRs, i.e., STRs with single base pair (bp) motifs (A, T, C, or G), can alter DNA accessibility[9] and contribute to dictating nucleosome organization[10–17]. In plant and human systems, genome studies have shown that intra-population length variation in STRs, including dimer STRs (two bp motifs; AT, CG, AG, AC, GT, and CT), correlates with expression variation in nearby genes[18–24]. A mechanistic explanation for the observed changes in gene expression is likely that sequence-specific transcription factors (TFs) are affected by local repetitiveness when binding DNA[25].

A key evolutionary aspect of STRs is their mutation rate that exceeds the rate of single nucleotide transversions and transitions[26],

and a non-random positioning of STRs would lead to non-random mutations along the genome sequence. Furthermore, genome-wide mutation accumulation studies in multiple systems (animals, plants, and fungi) have reported sequence context to be predictive of mutation rates[27–30]. Strikingly, the rate of insertions and deletions, which occur disproportionally in repetitive sequences, are modified by the external environment to a greater extent than single nucleotide transversions and transitions[31–33]. In bacterial systems, STRs that mediate stochastic switching of gene expression (on or off) tend to reside in adaptive genes linked to immune system evasion and surface adhesion[34,35], but whether the probability of STR-mutation is increased near certain types of eukaryote genes is not known.

Despite the indications that STRs play a role in tuning eukaryote gene regulation and thus impacting phenotypic traits, the fine-grained spatial occurrence of STRs and the predicted functions of nearby genes has not been examined systematically. To understand the

[1]Centre for Ecological and Evolutionary Synthesis, Department of Biosciences, University of Oslo, Oslo, Norway. [2]Section for Genetics and Evolutionary Biology, Department of Biosciences, University of Oslo, Oslo, Norway. ✉e-mail: w.b.reinar@ibv.uio.no; k.s.jakobsen@ibv.uio.no

distribution, function, and consequently the evolution of STRs in eukaryotes, we analyzed whole-genome data from 1270 eukaryotic species spanning seven eukaryotic supergroups. Fine-scaled analyzes of the monomer and dimer motif repetitiveness in ~5,1 billion gene-proximal regions revealed that fungal, animal, plant, algal, and other eukaryotic transcription and translation sites – i.e., across the eukaryotic Tree of Life (eToL) – were delineated by local shifts in monomer and dimer motif repetitiveness. We find that the repetitiveness landscape upstream and downstream of genes correlates with TF binding sites (TFBSs) and the functional category of the gene; genes with non-housekeeping functions had more dynamic repetitiveness profiles. Our results show that monomer and dimer STRs delineate gene bodies (exons and introns) and translation sites across the eToL and thus have evolved as a gene architectural trait in co-evolution with gene function and associated TFs. Given the hypermutability exerted by elevated repetitiveness, STRs likely contribute to the eco-evolutionary potential of all eukaryotic species.

## Results

### Monomer and dimer STRs delineate genes across the eukaryotic Tree of Life

The 1270 surveyed species encompassed members of seven eukaryotic supergroups: Obazoa ($n = 1067$), Amoebozoa ($n = 9$), excavates ($n = 21$), Cryptista ($n = 3$), Archaeplastida ($n = 82$), Haptista ($n = 1$), and Stramenopiles, Alveolata, and Rhizaria (SAR) ($n = 87$) (Fig. 1a). The genomes varied in size, GC-content, gene counts, and genome assembly quality metrics (Supplementary Fig. 1) and a species could be represented by multiple genome assemblies. Whole-genome scans of 2030 assemblies retrieved from Ensembl[36] resulted in an average of 4001 different non-coding STR motifs s per genome (Fig. 1b), but this varied from 28 STR motifs in the parasite *Encephalitozoon intestinalis* to 11,459 STR motifs in the grass *Aegilops tauschii*. Grouping motifs with their reverse and reverse complements the most common STR motifs in all groups were the monomer motifs A/T, followed by C/G, and either the dimer motif AC/CA/GT/TG in animals, AG/GA/TC/CT in fungi and plants and green algae (P&GA), and AT/TA in other eukaryotes (OE) (Fig. 1c). By comparing observed STR counts with randomly rearranged ("mock") STR datasets (*Methods*) we found that monomeric and dimeric STRs were strongly enriched within 100 bp upstream and downstream of gene annotation borders compared to STRs with larger unit sizes (i.e., trimers, tetramers, pentamers, and hexamers) (Fig. 1d). Indeed, monomer and dimer STR counts were elevated near the borders of annotated intergenic regions (Fig. 2a, b) regardless of intergenic length (Fig. 2c) but note that lacking or inconsistent annotation of 5′ and 3′ untranslated regions (UTRs) rendered the intergenic borders a mix of transcription and translation sites (Supplementary Fig. 2). This result shows that monomer and dimer STRs, on average, are found closer to the borders of intergenic regions than is expected by chance. To evaluate the gene-proximal repetitiveness of monomer and dimer motifs in detail, we calculated the number of monomer and dimer motifs succeeded or preceded by an identical motif in every 10-bp window 1000 bp upstream of translation start sites, i.e., upstream of the first start codon and 1000 bp downstream of translation stop sites, i.e., downstream of the last stop codon (Fig. 3a, b). To control for differences in background base composition we defined the deviation from the expected repetitiveness (henceforth $\Delta R_x$, where $_x$ indicates the motif) by subtracting the mean repetitiveness of the 1000-bp regions from the mean repetitiveness of the specific 10-bp windows (Fig. 3c–e). The mean genomic repetitiveness scores of all 10 monomeric and dimeric motifs (Fig. 4a) correlated significantly with the phylogenetic distances between species (all *P*-values < 0.001, Supplementary Table 1). In total, the exact repetitiveness scores of 5.1 billion windows were used to estimate the $\Delta R_x$ across the eToL. This produced 2000 $\Delta R_x$-scores per genome (100 windows upstream and 100 windows

downstream for each of the ten repeat motifs), including T-statistics and *P*-values capturing potential rejections of null hypotheses of equal means (see "*Methods*"). $\Delta R_x$ are thus continuous variables reflecting shifts in gene-proximal repetitiveness that controls for differences in background gene-proximal base compositions among the genomes of the surveyed organisms. The $\Delta R_x$-curve of one organism will only produce a clear spatial shape if enough of its genes contain similar patterning of gene-proximal repetitiveness. Further, we reason that the average $\Delta R_x$-scores across organism groups would only produce clear, spatial patterns if shifts in gene-proximal repetitiveness are conserved within the group in focus. By averaging the $\Delta R_x$-scores of animals, fungi, P&GA, and OE we observed conserved and abrupt gene-proximal repetitiveness shifts in a motif and group-specific manner (Fig. 4b). Interestingly, monomer $\Delta R_x$-scores ($\Delta R_A$, $\Delta R_T$, $\Delta R_C$, and $\Delta R_G$) from -1000 bp to the translation start site of animal genes broadly resembled those of P&GA while monomer $\Delta R_x$-scores upstream of fungal genes broadly resembled those of OE. Specifically, P&GA and animal genes displayed a steep decrease in $\Delta R_A$ followed by a steep increase close to the translation start site, whereas fungal and OE genes displayed a steady increase of $\Delta R_A$ towards the translation start site. $\Delta R_T$ decreased steadily and exhibited a sharp drop in P&GA, whereas $\Delta R_T$ slightly increased prior to a drop in animals, resembling the shape of $\Delta R_T$ upstream of fungal and OE genes. $\Delta R_C$ increased steadily toward animal, fungal, and OE translation start sites, whereas P&GA displayed a second peak of $\Delta R_C$ farther upstream. Conversely, P&GA $\Delta R_G$ sharply increased immediate to translation start sites, whereas animal genes displayed a second peak of $\Delta R_G$ farther upstream. In fungi and OE, $\Delta R_G$ steadily decreased towards the translation start site. From the translation stop site to +1000 bp, the $\Delta R_A$ of animal, P&GA, and OE resembled the upstream pattern of $\Delta R_A$, but did not exhibit strong spatial shifts in fungi. $\Delta R_T$ of P&GA steadily increased in fungi and OE prior to a drop near the translation stop, whereas the observed increase was slighter in animals. A drop of $\Delta R_C$ downstream of translation stop sites was evident in all groups except animals, while all groups shared a sharp increase of $\Delta R_G$ right upstream of translation stop sites. Of dimer motifs upstream of translation start sites, the strongest spatial shifts were exhibited by $\Delta R_{CT}$, which increased sharply just prior to translation in all groups, and $\Delta R_{AC}$, increasing towards the translation start in all groups except animals. Animals had a unique peak of $\Delta R_{CG}$ upstream of translation. $\Delta R_{AT}$ showed a bimodal pattern upstream of P&GA and fungal genes and steadily increased in OE genes but decreased towards the translation start in animals. $\Delta R_{AT}$ had the strongest shifts downstream of translation stop sites as well, clearly increasing towards translation stop sites in fungal and OE genes while decreasing in P&GA and animals. Pronounced peaks of $\Delta R_{GT}$ were present downstream of translation stop in all groups but animals. These results (Fig. 4a, b) show that repetitiveness and spatial shifts in repetitiveness are conserved features of eukaryotic genes across the eToL. Indeed, virtually all of the examined 1270 species had statistically significant shifts in $\Delta R$ (Supplementary Fig. 3). Clustering analysis (Principal Component; PC) of $\Delta R$-scores suggested an overall similarity between spatial $\Delta R$-scores in animal genomes and P&GA on the primary PC axis (Fig. 4c). Fungal species clustered together with OE on the primary PC axis, but fungal species were spread along the entire secondary axis, indicating high heterogeneity in $\Delta R$-scores, while most OE clustered together in the bottom left quadrant (Fig. 4c).

### Repetitiveness shifts are associated with translation and transcription sites

Shifts in repetitiveness were clearly linked to the location of translation start and stop sites (Fig. 4b). As stated, this analysis did not consider the location of transcription start and stop sites since UTR annotations were not present in all the surveyed genes (Supplementary Fig. 2). However, we investigated if shifts in $\Delta R$ coincided with transcription

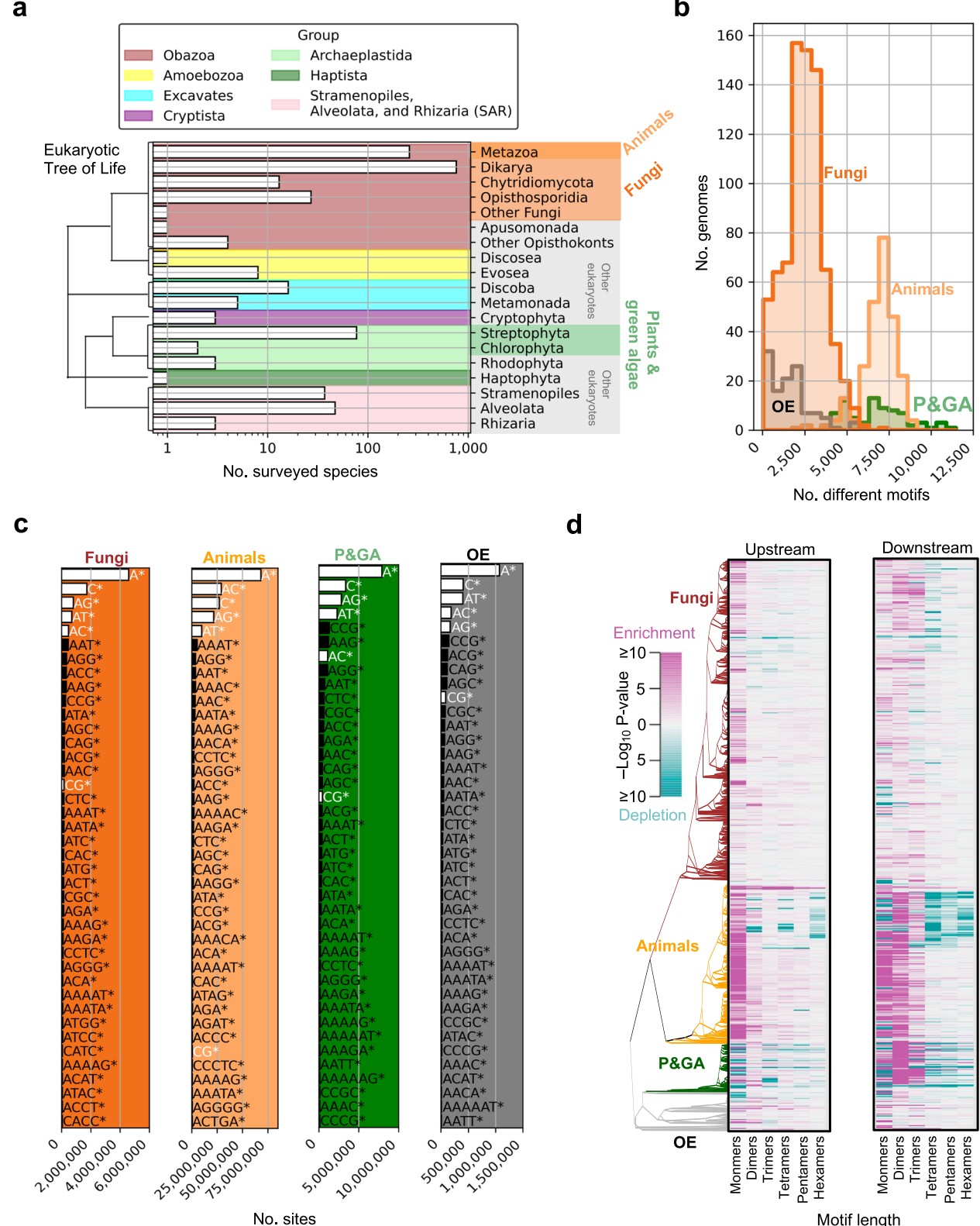

**Fig. 1 | Short tandem repeats (STRs) in eukaryotic species. a** The number of surveyed species. Animals, fungi, and plants and green algae are group designations used throughout this study. The remaining groups are referred to as "Other eukaryotes" (OE). The schematic eukaryotic Tree of Life follows Burki et al.[60]. Note the log-scaled *x*-axis. **b** The number of different STR motifs (*x*-axis) in animal, fungi, plants and green algae (P&GA), and other eukaryotic (OE) genomes. **c** The number of STR sites per motif in eukaryotic groups (the top 40 motifs are shown). The asterisks indicate that the shown motif, its reverse counterpart, and its reverse complement were grouped. Monomeric and dimeric motifs are the focus of this study and are highlighted with white bars and white text. **d** Two-sided Fisher's exact test *P*-values indicating enrichment (pink) or depletion (blue-green) of STR motifs within 100 bp upstream or downstream of annotated genes conditioned on unit length (x-axis: monomeric, dimeric, trimeric, tetrameric, pentameric, or hexameric). Note that (**d**) shows a subset of the 1270 surveyed species (891 species with phylogenetic data). Source data are provided as a Source Data file.

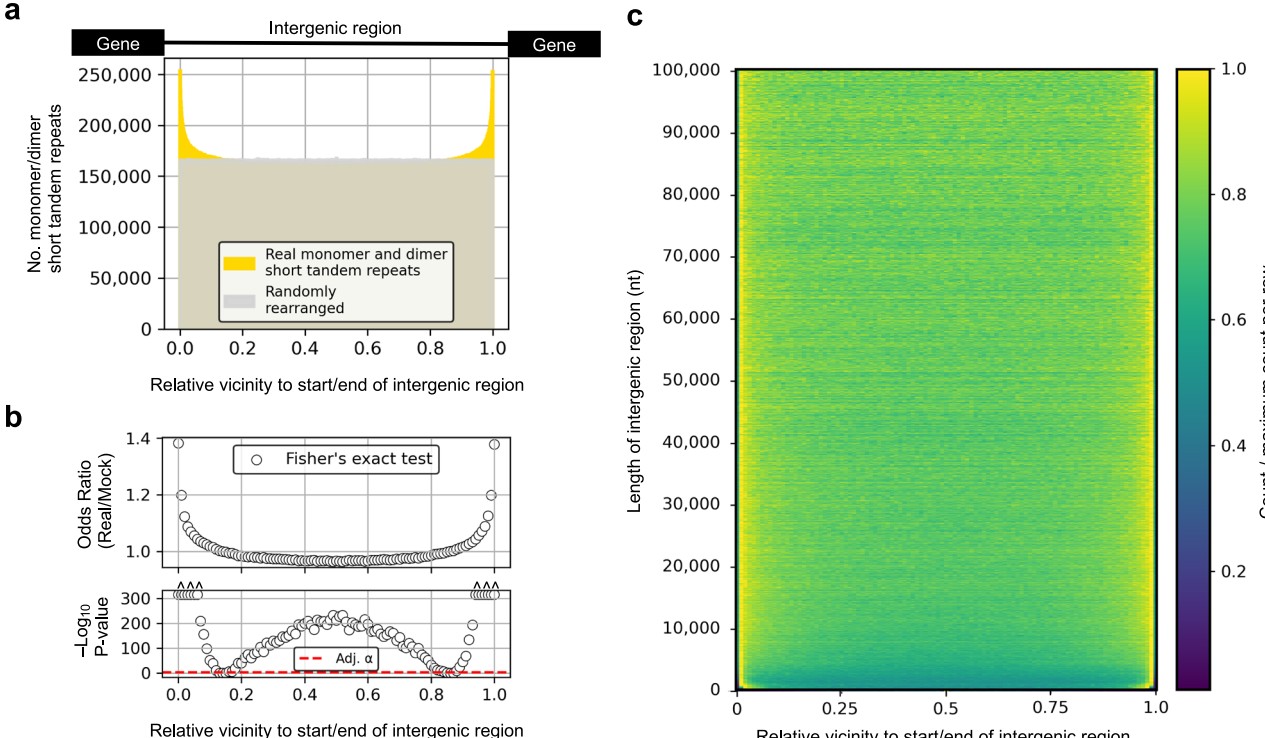

**Fig. 2 | Monomer and dimer short tandem repeat (STR) positions in intergenic sequences.** In all panels (**a**, **b**, and **c**), the data consists of 165,511,573 monomer STRs (i.e., STRs with A, T, C, and G as the repeated motif) and dimer STRs (i.e., STRs with AT, CG, AC, AG, CT, or GT as the repeated motif) in 11,390,652 intergenic regions detected in 2030 genome assemblies (1270 species). **a** The histograms indicate the number of monomer/dimer STRs (y-axis) as function of their position (x-axis) in intergenic regions (see cartoon). The yellow histogram indicates real positions whereas the gray histogram indicates the position of randomly rearranged ('mock' control) positions. The bin size was set to 1000. **b** Odds ratios (top panel) and *P*-values (bottom panel, red line indicates the Bonferroni adjusted α-threshold) from two-sided Fisher's exact tests of dependence between the relative vicinity to starts/ ends of intergenic regions and whether the data was real or randomly rearranged. **c** Per intergenic region size grouped in 100 bp intervals the STR counts as proportions of the maximum (colored from dark blue to yellow) are shown as a function of the relative vicinity to start/end of intergenic regions. Source data are provided as a Source Data file.

sites in the 1270 species by extracting the mean length of the 5' UTR and 3' UTR from the ENSEMBL gene annotations and investigated the presence of overlap between shifts in $\Delta R_x$ and the mean position of transcription start and stop sites. All groups had annotated transcription starts and stops that clearly overlapped with shifts in $\Delta R_x$ (Fig. 5). To formally quantify and assess the statistical significance of the patterns, we tested if the median sequence repetitiveness differed between sequences upstream of 5'UTR annotations and the 5'UTR sequences, and between sequences downstream of 3'UTR annotations and the 3'UTR sequences, as well as the region from the translation site to −50 bp upstream compared to the remaining upstream region (up to −1000 bp) and the region from the translation stop site to +50 bp downstream compared to the remaining downstream region (up to +1000 bp) (Fig. 6a). Overall, these tests supported that sequence within UTRs and near translation sites had distinct repetitiveness scores compared to nearby sequence, but the specific motif clearly depended on the taxa (Fig. 6b, Supplementary Fig. 4–5).

**Repetitiveness shifts are intensified in genomes with uniform base compositions**

The surveyed genomes had GC-contents ranging from < 20% to ~70% (Supplementary Fig. 1), and as expected, the repetitiveness of all motifs was correlated to the GC-contents of the genomes (Supplementary Fig. 6). Since our results indicated that shifts in repetitiveness demarcate transcription and translation (Figs. 5, 6), we assessed if repetitiveness shifts were intensified in genomes with more uniform base compositions. Interestingly, we found that $\Delta R_A$ was inflated near translation starts in GC-poor genomes, while $\Delta R_T$ decreased, and conversely, the $\Delta R_C$ intensified near translation sites in GC-rich

genomes, while $\Delta R_G$ decreased (Supplementary Fig. 7). With respect to dimer repeat motifs, $\Delta R_{AC}$ steeply increased upstream of the translation start sites in GC-rich genomes, $\Delta R_{CT}$ increased close to both translation start and stop sites, and $\Delta R_{AT}$ increased near translation stop sites in GC-poor genomes (Supplementary Fig. 8). Indeed, variation in gene-proximal repetitiveness of A, T, AT, C, G, CG, and AC correlated significantly with genome-wide GC-contents in phylogenetic generalized linear models (*P*-values < 0.005, Supplementary Fig. 9). These results suggest that the spatial patterns of repetitiveness have evolved in concert with mechanisms that drive the overall GC-content.

**Motif repetitiveness reflects functional categories of flanking genes**

While the patterns of repetitiveness deviations were conserved within studied eukaryotic groups, variations were observed at the level of individual genomes – i.e., genes within a genome displayed different local repetitiveness landscapes. We hypothesized that this variation in repetitiveness landscapes could reflect differences in gene function. To capture genes with the most pronounced shifts in repetitiveness, we ranked genes according to the standard deviation of their upstream and downstream repetitiveness (from -1000 bp to the translation start site and from the translation stop site to +1000 bp) and performed functional enrichment and depletion tests of the top 10% highest scoring genes per motif and per species (Fig. 7a, b). Next, we retrieved lists of housekeeping genes from Joshi et al.[37] covering 10 metazoan species (chimpanzee, gorilla, opossum, mouse, macaque, human, platypus, orangutan, *Caenorhabditis elegans*, and hamster) and their associated, enriched GO terms ("biological function") to produce a set

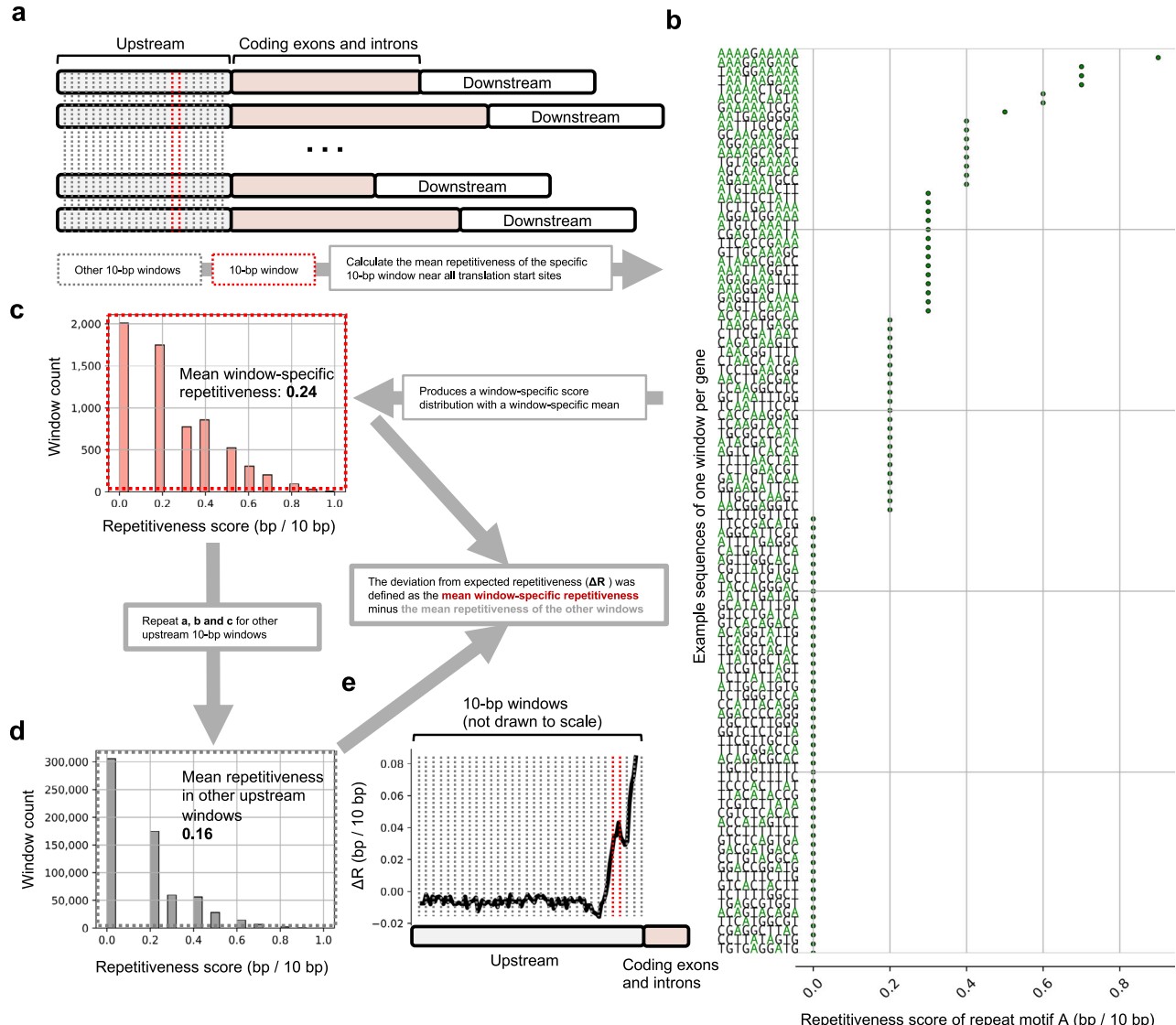

**Fig. 3 | Definitions of repetitiveness and ΔRₓ. a** For each 10-bp window in the 1000 bp region upstream and downstream of a gene we calculated the mean repetitiveness of the window as in **b** where repetitiveness was defined as the number of repeat motifs after or ensuing an identical repeat motif (shown here for the monomer repeat motif A). **c** The distribution of window-specific repetitiveness scores was used to calculate the mean window-specific repetitiveness and to test if the mean was significantly different from the region-specific mean shown in (**d**). **e** $\Delta R_x$ was defined as the mean window-specific repetitiveness minus the mean repetitiveness of the other windows and was calculated for each window in the 1000 bp regions upstream of translation start sites (depicted in the example graph in (**e**) and the gene cartoon in (**a**) and downstream of translation stop sites (not depicted).

---

of housekeeping functions. To test if shifts in gene-proximal repetitiveness were dependent on housekeeping functions, we compared the terms that were enriched or depleted in our gene lists with the terms linked to housekeeping. Genes with pronounced shifts in local repetitiveness were almost twice as likely to be depleted for housekeeping functions (Fisher's exact test odds ratio: 1.9, $P$-value $< 1.0 \times 10^{-308}$) (Fig. 7c). The statistical significance of the depletion for housekeeping functions varied along the eToL but was strongest in animals (Fig. 7d), likely reflecting a lack of functional studies in non-model systems, or a less defined division between genes involved in environmental responses and housekeeping in fungi, plants, and other eukaryotes. We conditioned the Fisher's exact test on phylogenetic groups and motifs and found that shifts in the repetitiveness of A, C, G, AT, CG, AC, AG, and CT were depleted near animal genes with housekeeping functions, whereas T and GT were enriched for genes with housekeeping functions (Supplementary Fig. 10a). In P&GA genomes, shifts in repetitiveness of A, T, AT, and AC were

depleted near genes with housekeeping functions, but shifts in G, AG, and GT were enriched (C, CG, and CT was not significant) (Supplementary Fig. 10b). In fungal genomes all tests gave non-significant results except depletions of T and AT near genes with housekeeping functions and an enrichment of G (Supplementary Fig. 10c). In other eukaryotes, the sole significant results were that T and AC were depleted near genes with housekeeping functions (Supplementary Fig. 10d). Thus, although the depletion and enrichment of specific motifs varied, we found differences in the local repetitiveness landscape when comparing genes with and without housekeeping functions across the major eukaryote groups. In species with available TFBS data through g:Profiler (fungi: *Saccharomyces cerevisiae*, nematodes: *Caenorhabditis elegans*, insects: *Drosophila melanogaster*, mammals: *Danio rerio, Homo sapiens, Mus musculus*, and *Pan troglodytes*) we scored enrichments of TFBS in the gene-proximal sequences of the above-mentioned high-scoring genes. In line with the recent discovery that a broad range of TFs bind or are affected by repeats in and around

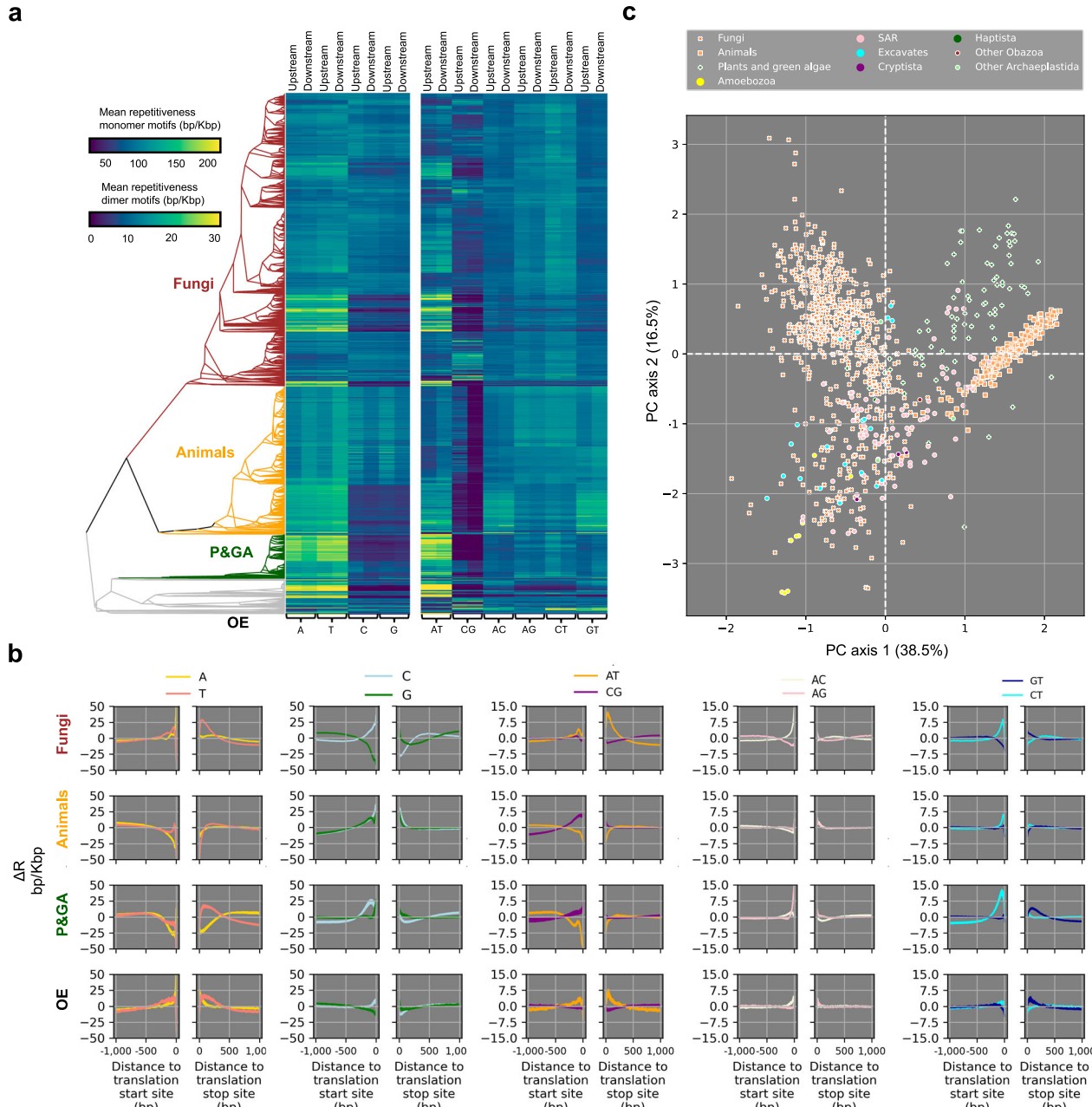

**Fig. 4 | Repetitiveness (R) and deviations from expected repetitiveness (ΔRₓ) across eukaryotes. a** The heatmap shows mean upstream and downstream *R*-values for monomer motifs (left panel) and dimer motifs (right panel) per motif and region (x-axes) and per species. Species are sorted phylogenetically and only species with phylogenetic data from TimeTree were included (891 species). **b** ΔRₓ-scores as a function of the distance to translation start sites and translation stop sites per monomer and dimer motif and per eukaryotic group. Each line shows the 95% confidence interval of the ΔRₓ-scores for one motif in 1475 fungal genome assemblies, 287 animal genome assemblies, 84 plant and green algae genome assemblies, and 201 other eukaryote genome assemblies. **c** Principal component (PC) analysis of ΔRₓ-scores. Each datapoint indicates the position of a species on PC axis 1 (*x*-axis) and PC axis 2 (*y*-axis). Species are colored by taxonomic group (see legend). Other Obazoa includes Apusomonada and other opisthokonts. Other Archaeplastida includes only Rhodophyta. Source data are provided as a Source Data file.

the TFBS[5], we found that gene-proximal regions with higher variation in repetitiveness of A, T, C, G, AT, CG, AC, and CT tended to have a TFBS containing the repeated motif across this broad assemblage of species (Fig. 7e, Supplementary Fig. 11). These results indicate that the local repetitiveness landscape of a gene correlates to the TF that regulates the gene and to the functional category of the gene.

## Discussion
The comprehensive genome-wide scans of short tandem repeats (STRs) reported here demonstrate that monomeric and dimeric STRs

are intrinsic to gene organization and delineate annotated gene bodies in animals, plants, fungi, and all other eukaryotic groups where whole genome assemblies exist (Figs. 1, 2). The detection of a STR site depends on the criteria used to define an STR, which vary among studies, and different definitions lead to different interpretations[38,39]. Further, the detection of STRs is sensitive to the heterogeneity in genome base compositions, captured efficiently by GC-content[40]. Thus, we used deviations from the expected repetitiveness, ΔRₓ, effectively controlling for base composition, to capture fine-grained patterns of STRs near genes. The fine-grained analyzes of motif

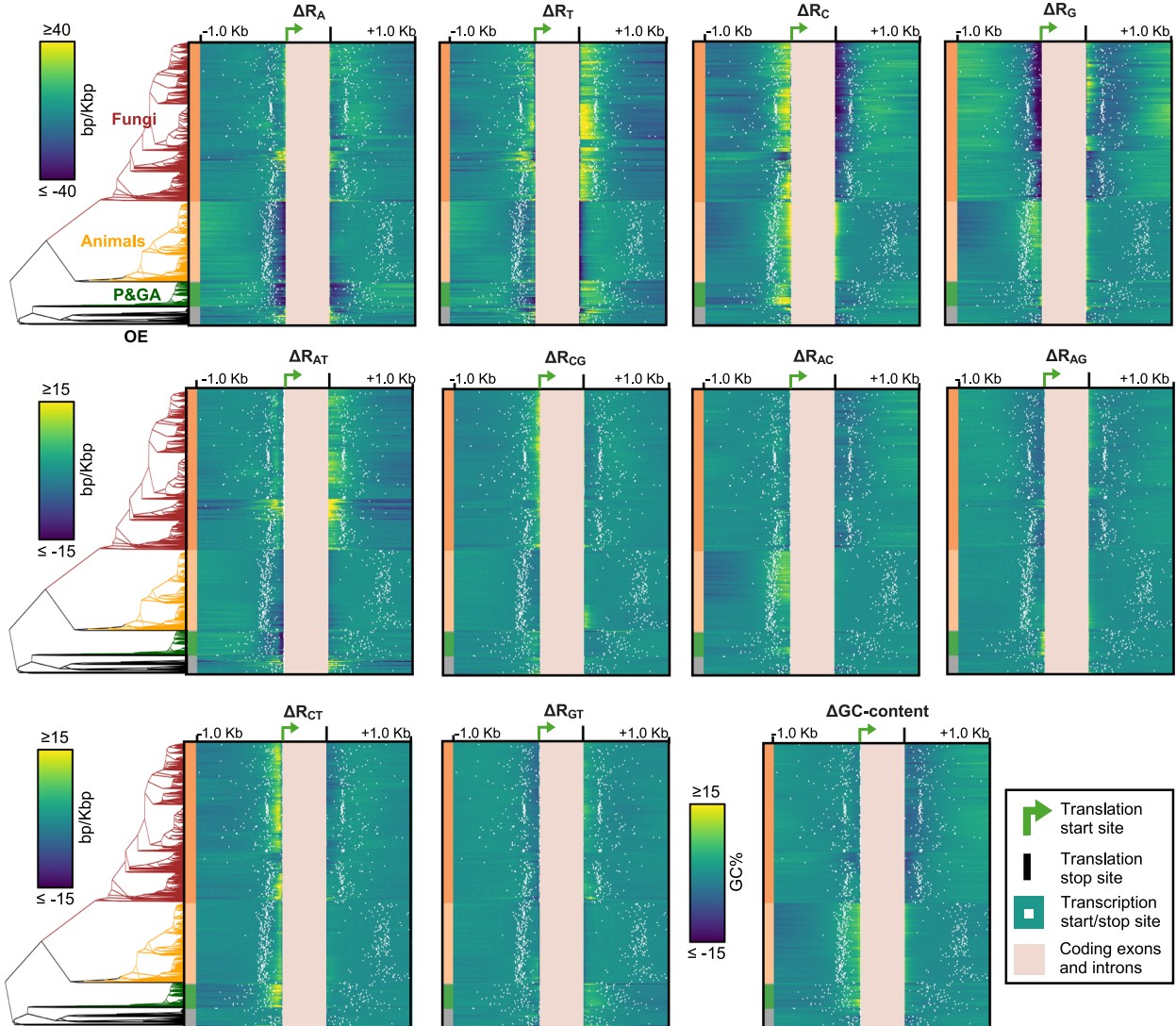

**Fig. 5 | Gene-proximal monomer and dimer repetitiveness in the context of transcription and translation across the eukaryotic Tree of Life.** The heatmaps indicate shifts of monomer repetitiveness ($\Delta R_A$, $\Delta R_T$, $\Delta R_C$, and $\Delta R_G$), dimer repetitiveness ($\Delta R_{AT}$, $\Delta R_{CG}$, $\Delta R_{AC}$, $\Delta R_{AG}$, $\Delta R_{CT}$, and $\Delta R_{GT}$), and shifts in GC-content ($\Delta GC$) in fungal, plant, animal and other eukaryotic species (see cladograms). Each row indicates one of the 891 different species and species scores were averaged if represented by multiple genome assemblies. The average positions of annotated transcription start and stop sites for each genome are indicated by white dots. For clarity, values were capped at −40 and 40 for monomer repetitiveness and −15 and 15 for dimer repetitiveness. Source data are provided as a Source Data file.

repetitiveness show that animals and green plants and algae differ from fungi and microeukaryotes in gene-proximal repetitiveness profiles (Fig. 4c), possibly due to genome size expansions and resulting regulatory and architectural requirements. Inter-species conservation of intergenic sequence structure has been linked to monomer and dimer STRs previously, including similarities in plants and animals[41], reflected by our detailed analyzes of monomer and dimer repetitiveness.

The mean sequence repetitiveness did not seem to reflect potential fixed positions of promoter motifs such as the TATA-box located approximately −30 upstream of many eukaryotic transcription start sites (Fig. 5), possibly due to the promoter variance among eukaryotic species in the presence and localization of regulatory elements. Although most disease-causing STRs are trimeric or larger[42] the frequent and consistent positioning of monomer and dimer STRs (Figs. 1d, 2) and monomer and dimer repetitiveness (Figs. 4–6) suggests functional roles across eukaryotes. Notably, species that have evolved extreme genomic base compositions, i.e., genomes that are

strongly biased by A/T (e.g. ~84% in the anaerobic gut fungi *Anaeromyces robustus*) or C/G (e.g. ~68% in the yeast *Tilletiopsis washingtonensis*), have intensified gene-proximal repetitiveness of their dominant base, corroborating that a local deviation in repetitiveness is a functional transcriptional and/or translational marker (Supplementary Figs. 7–9). Gene-proximal repetitiveness marked transcription and translation boundaries in a group-specific manner indicating that the repetitiveness around transcription and translation sites is conserved within taxa (Figs. 5, 6).

The high mutation rate of repetitive sites is due to frequent replication slippages[26]. The elevated mutability of repetitive sequences may explain why sequence complexity well predicts new mutations in many species (human[43], fruit fly[44], nematode[45], and green algae[46]). In well-studied systems, STR mutations have been proven to impact gene expression[2,21], splicing[47], and protein function[48]. It is therefore intriguing that variation in gene-proximal repetitiveness was lower near genes with housekeeping functions (Fig. 7c). This result aligns with the report that mutability correlates to gene

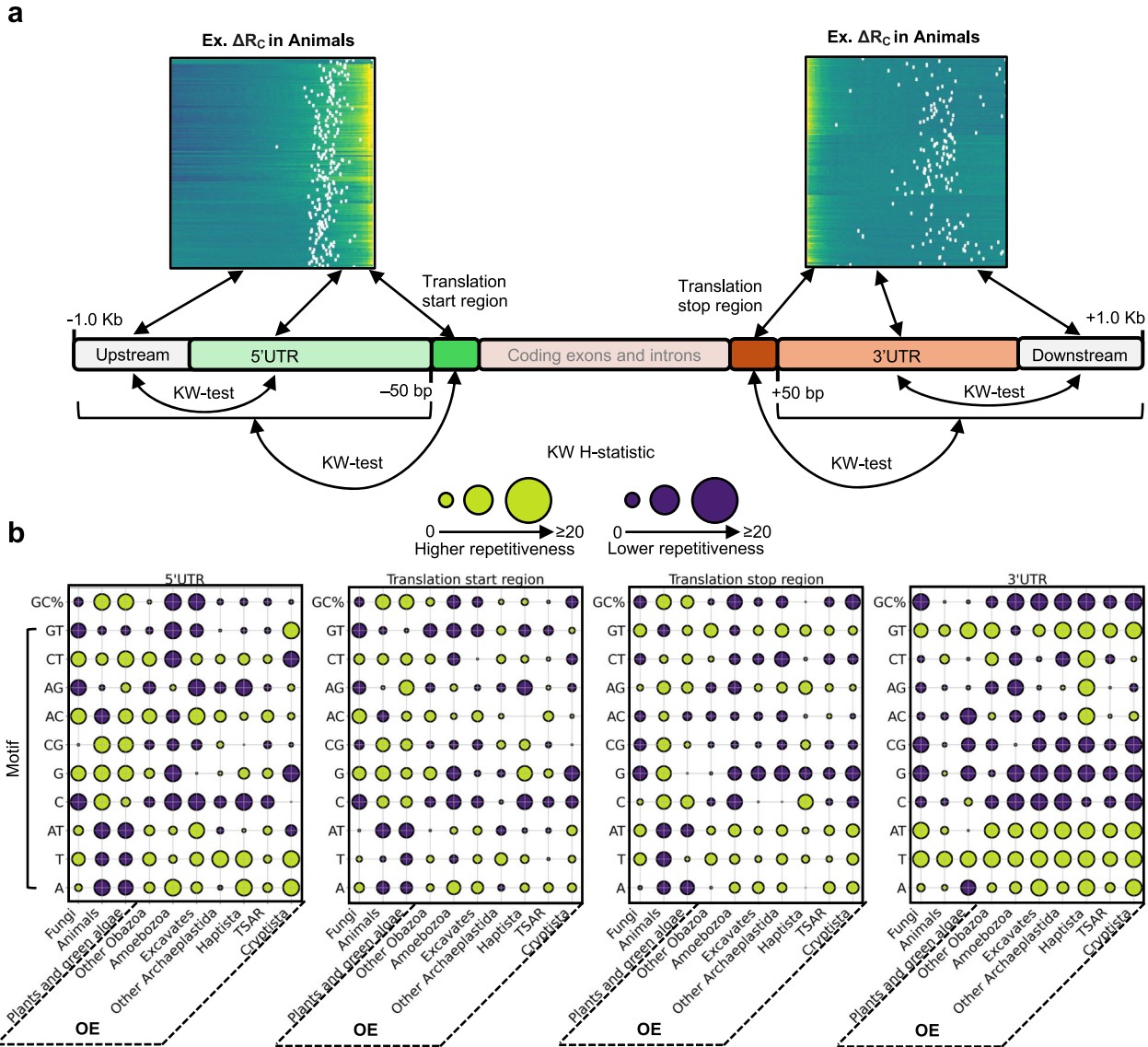

**Fig. 6 | Summary of repetitiveness shifts near transcription and translation across eukaryotic groups. a** The panels contain excerpts from Fig. 5, showing the distribution of $\Delta R_C$ in the gene-proximal regions of animal genomes. The cartoon (**a**) illustrates that the untranslated regions (5′ and 3′ UTRs) were compared to the upstream sequence up to −1000 bp (for 5′UTRs) or downstream sequences for (3′UTRs) up to +1000 bp with respect to the motif repetitiveness distribution in 10-bp windows using Kruskal-Wallis test (KW-test) of equal medians. Note that the size of the upstream and downstream regions depended on the species where the comparison was performed. Translation start regions were defined as the translation start site −50 bp and translation stop regions were defined as the translation stop site +50 bp. The distribution of repetitiveness scores in 10-bp windows, as well as the distribution of GC-content in 10 bp windows of the translation start and stop regions, were compared to the UTRs and the upstream (if 5′UTR) or downstream (if 3′UTR) region. **b** The KW-test results are shown for each comparison (5′UTR, translation start region, translation stop region, and 3′UTR) conditioned on eukaryotic group (x-axes) and motif (including GC-content) (y-axis). No. of species in each group: Fungi = 803, Animals = 259, Plants and green algae = 79, Other Obazoa = 5, Amoebozoa = 9, Excavates = 21, Other Archaeplastida = 3, Haptista = 1, TSAR = 87, and Cryptista = 3. Circles are scaled with the KW H-statistic so that larger blue circles indicate lower median repetitiveness and larger yellow circles indicate higher median repetitiveness in the region when compared to other sequences as indicated in (**a**). Exact KW H-statistics and *P*-values for each species are available in Supplementary Figs. 4–5. OE, Other eukaryotes. Source data are provided as a Source Data file.

function in *A. thaliana*[29] populations as we found that variation in gene-proximal motif repetitiveness is intertwined with gene function in plants and animals (Fig. 7d). The repetitiveness of the motifs may dictate the arsenal of transcription factors (TFs) that bind gene-proximal sequences to facilitate gene expression (Fig. 7e) and given that naturally occurring fitness-controlling variants disproportionally locate to nucleotides near or in TF binding sites (such as in *S. cerevisiae*[49]) it is likely that hypermutations in gene-proximal repeats near environmentally responsive genes strongly contribute to variation in fitness. The local repetitiveness landscapes of genes with housekeeping functions differed and may, therefore, be less exposed

to an evolutionary drive towards new regulatory mutations. On the other hand, the functional outcome of a deeply conserved plant gene has been influenced by mutations in proximal upstream and downstream sequence[50] indicating that the mutability of gene-proximal sequences near all genes has the potential to impact their regulatory outcome.

In conclusion, our findings demonstrate that local shifts in repetitiveness characterize eukaryotic genes and considering the known elevation of mutation rates in repetitive sites in fungi, animals, plants, and other eukaryotic groups, the shifts likely determine the probability of regulatory mutation. As such, local repetitiveness landscapes have

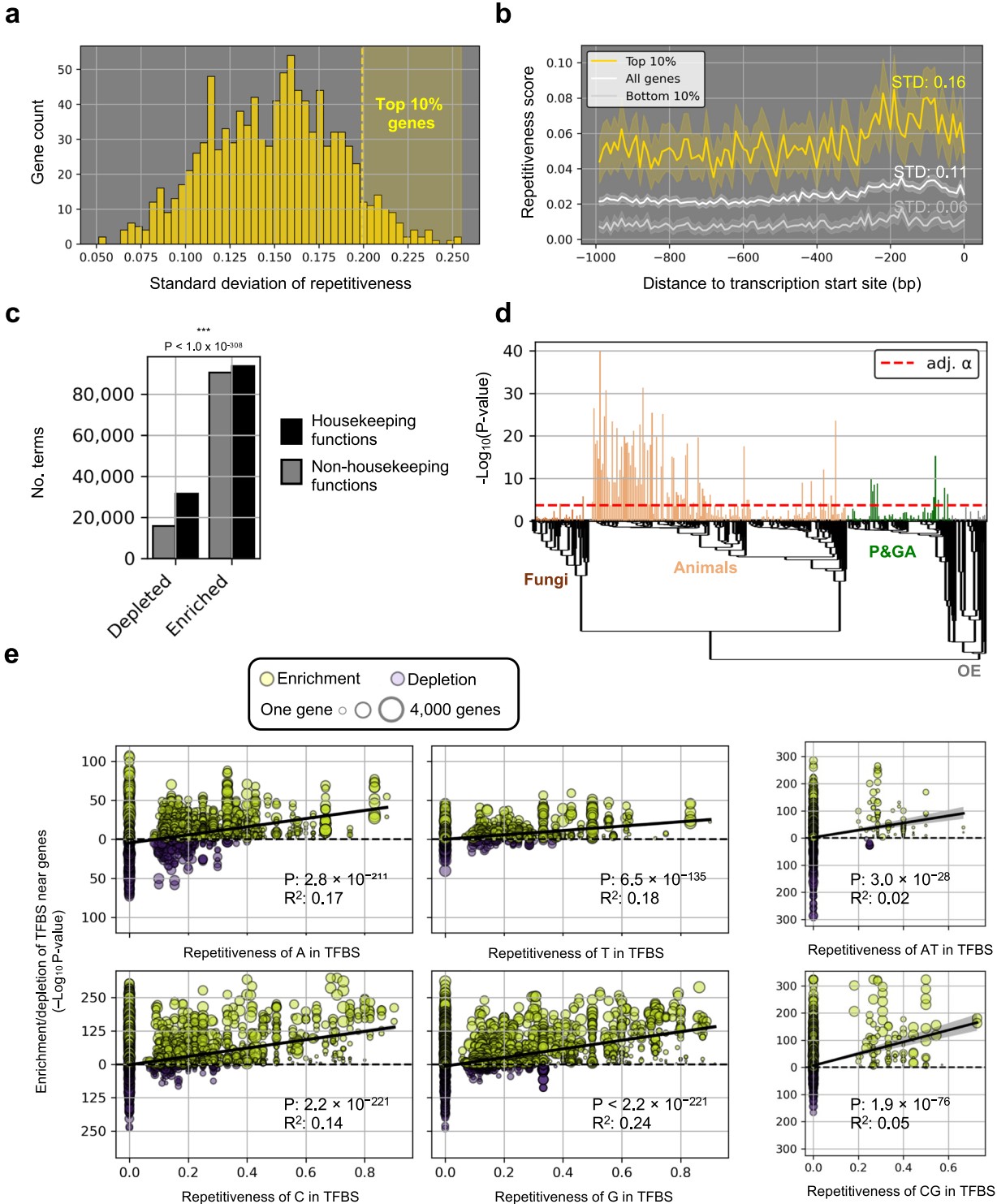

**Fig. 7 | Gene-proximal shifts in motif repetitiveness, gene function, and the transcription factor binding site (TFBS) landscape. a** The histogram shows example data (1000 human genes) serves to indicates how genes were chosen for gene ontology (GO) and TFBS analyzes: 10% of the top-scoring genes based on the variance in proximal repetitiveness were chosen for each genome assembly with available GO and TFBS data. **b** The lines show example data (1000 human genes) and indicate how variation in repetitiveness impacts the standard deviation score. The shaded areas indicate the 95% confidence interval of the mean. **c** The bar plot indicates the number of terms (dark gray: non-housekeeping functions, black: housekeeping functions) that were statistically depleted and enriched in the gene lists containing the top 10% genes in 316 species (171 animal species, 43 fungal species, 71 plants and green algae species, and 31 other eukaryote species). The *P*-value resulted from two-sided Fisher's exact test of dependence between

housekeeping functions and whether a term was depleted or enriched. **d** The result of two-sided Fisher's exact test per genome assembly along the eukaryotic phylogeny. The red, dashed line indicates the Bonferroni-adjusted α-threshold for multiple tests. **e** The *y*-axis indicate the depletion/enrichment scores of TFBS in intergenic regions flanking the top 10% high-scoring genes in seven species, using hypergeometric tests separately for enrichment and depletion, with species-level g:SCS correction for multiple tests. The enrichment/depletion *P*-values are shown as a function of the repetitiveness of the TFBS. Circle sizes are scaled with the number of genes associated with the TFBS. Yellow color indicates a statistically significant enrichment, and purple color indicates a statistically significant depletion. The $R^2$ and *P*-value result from a linear fit and the shaded area indicate the 95% confidence interval for the regression estimate. Source data are provided as a Source Data file.

co-evolved with gene function due to negative selection for repetitiveness near genes with housekeeping functions.

## Methods

### Data and metadata sources

Genome FASTA files and general feature format (GFF) files available from Ensembl[36] release 105, EnsemblPlants release 53, EnsemblFungi release 52, and EnsemblProtists release 53 were downloaded from https://ftp.ensembl.org/pub/. We queried the Genomes on a Tree[51] (GoaT; https://goat.genomehubs.org/) meta-database and mapped the "assembly_id" column to the Ensembl assembly accession codes of the downloaded genome assemblies to retrieve genome assembly metadata (columns: "taxon_id", "taxon_rank", "scientific_name", "assembly_id", "assembly_level", "assembly_span", "chromosome_count", "contig_count", "contig_n50", "gene_count", "scaffold_count", "scaffold_n50", "gc_percent", "n_percent", and "busco_completeness"). From a second query to the GoaT meta-database we retrieved taxonomic designations (columns: "taxon_id", "taxon_rank", "scientific_name", "subspecies", "species", "genus", "family", "order", "class", "phylum", "kingdom", "superkingdom") for each genome assembly. The taxonomic designations were manually curated to produce ranks not necessarily included in the taxonomy derived by GoaT, including the categorizations presented in Fig. 1. Genome assembly metadata was manually checked, and only one obvious mistake was present: the "gene_count" value for assembly GCA_902713445.1 was unrealistically high (2,1 million genes) and was set to the number of protein-coding genes (55,174) as stated on the NCBI website (https://www.ncbi.nlm.nih.gov/datasets/genome/GCA_902713445.1/). The phylogenetic tree used for visualization and statistics was retrieved from TimeTree[52].

### Spatial analysis of STRs in whole-genome assemblies

ULTRA Locates Tandemly Repetitive Areas (ULTRA) v.0.99.17 was run with standard parameters on the FASTA files retrieved from Ensembl. ULTRA detects tandem repeats as described in ref. 53. The JSON files produced by ULTRA were parsed with Python code to produce a PyRanges[54] (v.0.0.117) object. Genic regions were defined as regions in the GFF files annotated as "gene" in the "Feature" column. A PyRanges object containing the chromosome coordinates was used to retrieve the intergenic coordinates (by use of the PyRanges "subtract"-function), and the "join"-function was used with the "how = containment" parameter to identify STRs within intergenic regions. The midpoint of an STR was defined as the STR start coordinate plus half its length. For each STR, we created a mock version of the STR that was randomly placed within its respective intergenic region by using the "random.randint"-function of NumPy[55] (v.1.21.5) to generate a random start coordinate, and the end coordinate was set as the start coordinate plus the STR length. We counted the number of nucleotides from the STR midpoint to the closest nucleotide present in an annotated "gene" region and the length of the intergenic region. The number of nucleotides from the STR midpoint divided by the length of the intergenic region yielded values ranging from 0 to 1.0, where low and high values indicate relative closeness to the intergenic borders.

### Enrichment/depletion of STRs within 100 bp of gene regions

We compared the distribution of randomly rearranged STRs (see "Methods" section "Spatial analysis of STRs in whole-genome assemblies") with the observed distribution of STRs by performing two-sided Fisher's exact tests specifically testing if the counts of STRs of a given unit size were more often found within 100 bp compared to their randomly rearranged (mock) counterparts. The ratio of real counts$_{within\ 100\ bp}$ / real counts$_{outside\ 100\ bp}$ was divided by mock counts$_{within\ 100\ bp}$ / mock outside$_{100\ bp}$ to produce odds ratios and $P$-

values. One test was performed per genome assembly and $P$-values were adjusted for multiple testing. The Fisher's exact test was used as implemented by the "fisher_exact" function of SciPy[56] v.1.12.0.

### Deviations from expected STR counts

Observed monomer and dimer STR counts were compared to their mock counterparts per relative distance unit, i.e., the proportion of the intergenic region relative to the closest gene annotation, by using two-sided Fisher's exact test as implemented in SciPy v.1.12.0: The ratio of real counts$_{in\ proportion}$/ real counts$_{not\ in\ proportion}$ was divided by the ratio of mock counts$_{in\ proportion}$/ mock counts$_{not\ in\ proportion}$ to retrieve the odds ratio and the $P$-value every interval from 0 to 1.0 with a step size equaling 0.01.

### Spatial analysis of repeat motif repetitiveness

We selected one gene transcript among different gene isoforms by calculating the length of regions annotated as "mRNA" in the Ensembl GFFs, sorted the regions by length and chose the longest transcript for analysis. We located the coding DNA sequence (CDS) regions belonging to the longest transcript per gene and extracted the start coordinate of the first bp of the CDS and the end coordinate of the last bp of the CDS. These coordinates were considered the translation start and stop site, respectively. The pipeline simultaneously calculated the mean length of the 5′ and 3′ untranslated region (UTR) if present for the transcript. Supplementary Fig. 2 shows the ratio of genes with UTR annotations in each genome assembly. Next, we extracted the complete chromosome for each transcript where it resided, as a Python text string (using pyfastx[57] v.0.8.4). Considering strandedness, we extracted 1000 bp upstream of the first CDS and 1000 bp downstream of the last CDS. Per 10-bp window in the 1000 bp regions, we calculated the repetitiveness of each window using the Python script repetitiveness.py (see Code Availability). In the sliding-window analyzes, the window size of 10 bp and the step size of 10 were chosen as it balanced the required computational time and memory usage with an acceptable resolution. The Python script takes three inputs, m, w, and s, where m is the repeat motif, w is the window size and s is the sequence text string, and returns the repetitiveness scores, which is the number of repeat motifs found in a text string that follows or is superseded by an identical repeat motif, divided the length of the input text string (Fig. 3a, b). To calculate GC-content, the GC-content of the text string was calculated by counting the number of Gs and Cs in the sequence and dividing the total count by the length of the sequence.

### Definition of $\Delta R_x$ and T-tests of equal mean repetitiveness

$\Delta R_x$ was defined on a genome per genome basis as the difference between the mean repetitiveness of a window compared to the mean repetitiveness of all other windows (Fig. 3c–e). The repetitiveness scores for each window 1000 bp upstream of the translation start site and 1000 bp downstream of the translation stop site were used to identify statistically significant deviations in repetitiveness. Separately for upstream and downstream regions, and separately per motif, we tested the null hypothesis that the mean repetitiveness score of one window was equal to the mean repetitiveness score of all other windows in the specific genome using two-sided T-tests implemented in the "ttest_ind"-function of SciPy v.1.7.3. We used the T-statistics and $P$-values resulting from the T-tests to evaluate the strength of evidence for rejecting the null hypotheses of equal mean repetitiveness and equal mean GC-content. The sample size of the two-sided T-test depended on the number of analyzed genes in each assembly and ranged from 342 genes to 50,678 genes as indicated in the Source Data of Fig. 5. The degrees of freedom for each test equaled the number of windows (100) multiplied by the number of genes minus two.

## Principal Component Analysis (PCA) of repetitiveness

Based on the $\Delta R_x$ of each motif in each window per species, we constructed a matrix with the species as rows and repetitiveness conditioned on region (upstream or downstream) and motif (A, T, C, G, AT, CG, AC, AG, CT, GT) as values. The $1270 \times 2000$ matrix was decomposed by the "decomposition.PCA"-function of scikit-learn[58] with the parameter "n_components" set to 4. When species were represented with data from multiple genome assemblies, mean values were used.

## GO and TFBS enrichment/depletion

The rationale for this analysis was to identify the predicted function of genes with peaks and dips in repetitiveness proximal to translation start sites. In principle, we could have ranked the genes by the mean gene-proximal repetitiveness, but we would risk a bias towards genes in genomic regions with more uniform base compositions and thus elevated repetitiveness by default. To focus on genes with high local deviations in proximal repetitiveness compared to other genes, we calculated the standard deviation (STD) of the upstream and downstream score distributions conditioned on gene and repeat motif and used the NumPy "percentile"-function to extract gene identifiers above or in the 90th percentile when ranked by the STD of repetitiveness, i.e., genes with high variation in repetitiveness. By this procedure, we produced lists per genome of the gene identifiers above or in the 90th percentile. These gene lists were queried for enrichment and depletion of GO terms ("biological function") and TFs based on the hypergeometric probability function implemented in g:Profiler[59] with the standard domain scope (all annotated genes). Multiple test correction was performed by the g:SCS algorithm; see ref. 59 for further descriptions of the statistical test. To retrieve terms with housekeeping functions we extracted gene lists from ref. 37 and used g:Profiler to retrieve statistically enriched terms. The statistical significance of dependence between terms with housekeeping functions and enriched/depleted terms were assessed using two-sided Fisher's exact tests per motif: The ratio of housekeeping terms$_{depleted}$ / housekeeping terms$_{enriched}$ was divided by non-housekeeping terms$_{depleted}$ / non-housekeeping terms$_{enriched}$ to produce odds ratios and $P$-values.

## Reporting summary

Further information on research design is available in the Nature Portfolio Reporting Summary linked to this article.

## Data availability

Ensembl genome assemblies and gene annotations used in this study are publicly available at https://ftp.ensemblgenomes.ebi.ac.uk/pub/plants/, https://ftp.ensemblgenomes.ebi.ac.uk/pub/protists/, https://ftp.ensemblgenomes.ebi.ac.uk/pub/fungi/, and https://ftp.ensembl.org/pub/. Source data can be found in the Figshare Repository (https://doi.org/10.6084/m9.figshare.25427812). Source data are provided with this paper.

## Code availability

Code used in this study is available in the Figshare Repository (https://doi.org/10.6084/m9.figshare.25427812).

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

## Acknowledgements

The computational work in this study was performed on the Saga supercomputing clusters operated by the Research Computing Services group at USIT, UNINETT Sigma2, and the University of Oslo IT department. The work was funded by the University of Oslo (W.B.R and V.O.L) and by RCN Grant 251076 to K.S.J and M.A.B.

## Author contributions

K.S.J and M.A.B. conceived the project. W.B.R. analyzed the data and wrote the manuscript with input from K.S.J, M.A.B, V.O.L, and A.K.K. A.K.K. curated the taxonomy.

## Competing interests

The authors declare no competing interests.
