## [Transparent Peer Review file · Nature Communications]

Short tandem repeats delineate gene bodies across eukaryotes

Corresponding Author: Dr William Reinar

Version 0:

Reviewer comments:

Reviewer #1

(Remarks to the Author)

Understanding forces shaping the evolution of genome architecture is a longstanding project for biology, and in their manuscript, Reinar et al provide compelling and novel insights into the evolution of repetitive sequences of neighboring genes. I was impressed by the scale and creativity of the analyses performed, which revealed a number of remarkable patterns across organisms. For example, the presence of repeats of neighboring genes appears to be a common feature, but the sequence composition of repeats found in gene boundaries can vary among taxa across the tree of life. The finding that these repeat features are linked closely to gene functions (i.e. essential vs environmentally responsive) was also revealing. The paper is well written and I think this will make a valuable publication, but I do have a few minor comments and one significant comment.

Line 15: 5,1 billion (is this 5.1?) elsewhere 5.1 billion is used, but “5,1” shows up a few times.

Figure 2b. I know it's in the legend, but labeling this as a “randomization” directly in the plot might be helpful for readers.

Fig.3 As nice looking as this is, it's hard to figure out where on the x-axis and y-axis the data line up with the 3-D view. I wonder if stacked panels for each taxa would be a clearer visualization. I think Extended Data Fig. 4/5 are close, but I imagine you could plot multiple lines on the same plot or similar so it can fit as a main text figure. (leave it to authors/editor, to decide if this is essential)

Figure 5. Please label the subpanels with which nucleotide/dinucleotide is being plotted. It is tedious to scan back and forth from the legend to know what you are looking at in each panel.

Some discussion from the authors on what mechanisms (DNA repair, natural selection, etc) might be responsible for the enrichment of certain repeats in relation to genes across taxa would be interesting.

Major: I was surprised the authors have not done any formal analyses or visualizations of the phylogenetic context of the patterns they are finding. I think adding something along these lines would enhance support for some of the conclusions. The authors should look for phylogenetic signal of various features (ie. Rx scores of specific nucleotides, etc) explicitly. For example see ape package in R, which have methods to test for phylogenetic signals, correlated evolution of traits, ancestral state reconstruction, etc. A time-calibrated phylogenetic tree for the species could be obtained from TimeTree 5. Formally testing for correlated evolution would give more credence to statements like Line 186: “These results show that the spatial patterns of repetitiveness have evolved in concert with mechanisms that drive the overall GC-content.” While the results seem to suggest this, co-evolution of repetitiveness and GC-content can be formally tested in the context of a phylogeny. I think such analyses could be used in a number of places throughout this work and would provide statistical support for various comparisons across taxa. If this is new to the authors, I recommend “Analysis of Phylogenetics and Evolution with R”

(Remarks on code availability)

I tried to access the code the authors pointed to in their paper at <https://figshare.com/s/0788da931cf2de7e3cc6>

Unfortunately, I received this when trying to view the code file:

```
{"message": "Insufficient permissions", "code": "InsufficientPermissions"}
```

Reviewer #2

(Remarks to the Author)

This manuscript reports a so-far missing analysis on short tandem repeats (STRs) around genes across eukaryotes. Authors have some exciting results: that STRs are more abundant than expected close to genes and match to transcription factor binding sites. Yet there are certain aspects that need to be clarified so that the manuscript's value is good for publication.

Overall, I found the results of the TF analysis really interesting. If repetitiveness shifts around genes hold after showing 95% CI's, that's also great. I have multiple points regarding the GO Analysis. In general, it is not an easy read, because results often refer to extended figures and analyses are mostly not simple but then either they are not really backed up by stats or not clear how or why they are done the way they are done.

I did not find the manuscript ready for publication. I would love fellow repeat people publish their stuff in good journals. But there is some more work here. Hope this helps.

Major Points:

1. Using the name STR:

STRs vary between 1-6 nucleotides in units. Sometimes monomers act differently than others. So just analyzing mono- and dimers is hardly a paper on STRs. I would suggest being more specific since it is misleading otherwise. Also, it wasn't clearly stated why authors did not include longer units of STRs.

2. Statistics

Author use some indirect way to measure presence of repeat units around genes. They call the measure repetitiveness. It is basically comparing the extent of repeating elements in 10 nucleotide long windows to the rest for each of upstream and downstream region. Unfortunately, the statistical tests are sometimes missing. For example, the biggest claim that the authors make (and I think it is true but we need to show it correctly) is that there are more repeats closer to transcription/translation start/end sites. But then there should be a statistics associated with this claim. And that is not clear. I list all such claims below:

- line 65: Since you say the genomes varied, I would like to see some statistics test results for the Extended Data Figure 1. Maybe report it just below in the caption or as stars in the plot but a reporting of significance would be necessary.

- Extended Data Fig 4 and 5: All variation is plotted in 1 SD. It needs to be 2SD or 95% CI. I don't know if the patterns then hold. This I found worrisome, to be honest. Because if the 95% CI are too wide, then this would not be a helpful way to depict the varying abundance of repeats.

- Extended Data Figure 6: results are not corrected for multiple testing. Authors should then find another statistically sound way if correcting for too many tests here means too little power. There needs to be a correction. Otherwise no results can be deduced.

- Fig 6: '[if there are multiple assemblies per species, a random one is used.]' But for other analyses, average was taken. Why not here?

- GO Analysis: where are the statistical test results? Maybe I missed it. Based on the figure, I would not argue there are differences. Not obvious.

- GO Analysis: line 494-495: g profiler is used to test, all good but which statistical test was used, was a correction applied? these need to be described.

- GO Analysis: Extended Data Fig 9: this analysis is not helpful. I honestly did not understand, what we need to get out of it. And quantitatively, what it shows.

3. Clarity

Since the approaches authors take are not simple counts etc, I think clarity plays an important role. And there are certain things that are missing, which makes the text unclear. In conclusion for example, there are two points, where results are mentioned but no analysis are shown.

- Fig 2: Why has b higher counts than a? Because of the normalization? I think the amount of counts should be kept more even to convey the message.

- Extended Data Fig 11: line 412-413: Why is this normalization used? What does it exactly do? Is it responsible of the effect I asked about above?

- Transcription Factor Binding Analysis: why is this analysis done only on seven species?

- Conclusion, lines 252-253: this was not shown. This would be a great analysis that I suggest authors do though. Compare the GO's for high variation genes with housekeeping genes. This would yield a significant result for sure and it would be an important discovery. I would prefer that result over what we see on Fig 7a since this would be more clear and statistically backed up.

- Conclusion: line 256-257: how exactly does it relate to Fig 7b? Is there a stretch regarding the chromatin structure or did I miss an analysis?

- Methods: following questions would help with basic clarity:

How many STRs per genome are detected? how many are conserved?

What is GoaT exactly? Alignments?

Why is ULTRA used but not others? TRF, Tral etc.

Why not longer repeat units?

Minors:

- line 75-76, move the (Extended Data Figure 2) up in the sentence maybe after (UTRs) since it describes the missing data to be more precise.

- Fig 6: same color are used for species and variable, which becomes confusing. (maybe color only based on species?) The way how scaling is explained in the caption can be clearer.

- Extended Data Figure 7: Each line is what? Please mention it in the caption.

Further suggestions:

- Fig 5: maybe use more contrasting colors. That would help to convey the message. And make the white dots bigger? Hard to see them, too.

- Abstract: For the first two refs, which describe emergence of STRs' role on gene expression in genomes, the following first genome' wide publications would be more accurate: ours: Bilgin Sonay and Carvalho, 2015 and our collaborators: Willems et al., 2016

- Extended Data Fig 3: very helpful figure. I would consider putting this figure in the main text since it is hard to follow how R is calculated.

(Remarks on code availability)

Reviewer #3

(Remarks to the Author)

Reinar et al report a study of mono- and di-nucleotide repeat variation across a large number of taxa.

Major:

1. The definition of STRs in the manuscript needs clarification. The study is limited to repeating mono and dinucleotide sequences as explained in the Methods, but there does not appear to be a statement as to what constitutes a minimum size of a mononucleotide STR. Reading through the manuscript it is not clear whether the sequences 'AA' or 'AAA' might be identified as a STR in this work. The wording of the manuscript should be updated throughout to clarify this, and some justification as to this choice would be useful - the STRs may not reflect TF binding site lengths, and most (if not all) disease-causing STR expansions are trinucleotide or greater, for example, and it would be useful to see discussion of this.

2. The methodology in this version of the manuscript is unclear. The spatial analysis of whole genomes section appears to describe identification of the position of mononucleotide and dinucleotide sequences with ULTRA, then the positions are then randomly assigned to the same intergenic region in a 'mock STR' whose purpose is unclear - the word 'mock' is never mentioned again. The 'normalise_axis' parameter of Vaex is used, but the actual function is never described; what normalisation is performed? Are the normalised positions used elsewhere?

3. The authors propose a metric of repetitiveness that requires significant elaboration and is not well-explained in the Python notebook. Consider the sequences 'ACACACACAC' and 'ACACATACAC', which differ by a C->T substitution at one position. It would appear that the latter would be considered to be twice as repetitive as the former by merit of containing two AC runs, despite the sequence differing by only one nucleotide. Examples like 'AAAAAAAAA' vs 'AATAATAAT' show similar behaviour, with the latter apparently being labelled three times as repetitive as the first.

4. Fig 3. is difficult to read. It shows some changes between clades, but there is no indication of the significance of such changes and the use of 3D projections make it difficult to read the figure. Some comment as to what is happening here in the

context of known biology would be useful but is absent - for example, what is happening with TATA box elements, and is there a corresponding AT signal? Would these be considered STRs?

5. Fig 5. is extremely dense and would be clearer if reworked. The sites also appear to have been independently hierarchically clustered, which scrambles the orders across plots - as a result, it's not possible to see whether the changes in repetitiveness in mononucleotide A STRs are the inverse of those in mononucleotide T STRs.

6. Insufficient detail is provided for training of the gradient-boosted trees. The prediction targets should be properly defined in text, not solely in Fig 6, and any imbalance in target classes discussed along with strategies for addressing these. Hyperparameter selection should be explained. Calculation of robustness of predictor performance ideally would use a clearly described cross-validation strategy or a totally held-out subset of data rather than a reused fold of cross-validation.

7. Figure 7a appears to show no signal or discriminatory capacity on the axes - word clouds are not a good choice for visualisation here, and the use of too high a level of the ontology means that all axes appear to have 'response', or 'regulation', the meaning of which is unclear in the context of this plot. 7b appears to show that 'AT' presence is correlated in the opposite direction to 'CG' presence for CG-rich TF binding sites, which would appear to be a truism, but this is not commented on.

8. I would ideally like to see a more-thorough review of the existing literature of STR spatial distribution and more in-depth contextualisation of the key results here in the context of that body of work. The manuscript would also benefit from a clear discussion of how known features (RNA Pol II binding site motifs, for example) show in the metrics the authors define.

9. Additional detail is needed for the enrichment analysis parameters and the specific background gene lists used should be listed, and reducing the dimensionality to two dimensions is potentially oversimplifying the data - an approach with less reduction that selects the PCs explaining the majority of variation may be more appropriate.

Minor:

1. Commas are used in place of decimal points in numbers throughout text.
2. Extended Data Table 1 and Extended Data Table 2 do not contain data; they should probably be listed as a supplementary figure that shows the structure of the data, not as data itself.
3. Extended Figure 11 is a solid blue box.
4. Extended Figure 12 features illegible text at screen resolution.

(Remarks on code availability)

I manually reviewed but did not install and run the code. I have reservations about the appropriateness of the repetitiveness metric and would like to see this explained in text. No README file is provided, but I expect that the Python notebook would execute from manual review.

Version 1:

Reviewer comments:

Reviewer #1

(Remarks to the Author)

The authors have fully addressed all of my comments. My only remaining note is to encourage the authors to make sure the figure font sizes are legible per journal requirements (minimum size 6, I believe?)

I was also asked to comment on the authors response to Reviewer #2. While I think the reviewer has more expertise in satellite/repeat evolution in general than myself, I found the authors responses and revisions valuable. In particular, they have significantly improved the statistical rigor of their study, which was a primary concern from Reviewer #2. I find their responses acceptable and do not have further suggestions with respect to reviewer #2's comments.

(Remarks on code availability)

The authors provide a Jupyter notebook for running code and the datasets.

Reviewer #3

(Remarks to the Author)

Thank you for your revisions - nearly all of my points have been excellently addressed. Congratulations on a very nice paper! There is one minor change that I feel should be addressed further.

Regarding the response to point 8 of "Regarding the spatial distribution of STRs very little is known, which was the major rationale for this work.":

There is a moderate body of literature relating to the inter-species conservation and differences in length and spatial distribution of these elements that should be contextualised here. This appears in the literature as STRs, microsatellites, and sometimes k-mers.

Some examples that might be useful to include are:

<https://bmcgenomics.biomedcentral.com/articles/10.1186/s12864-019-5516-5>
<https://www.ncbi.nlm.nih.gov/pmc/articles/PMC5054066/#bib36>
<https://pubmed.ncbi.nlm.nih.gov/36289560/>
<https://www.mdpi.com/2073-4425/12/10/1571>
<https://genome.cshlp.org/content/13/10/2242>
<https://www.nature.com/articles/ng0795-337>

(Remarks on code availability)

Dear Editor,

Thank you for allowing us to submit a revised version of the manuscript.

We thank the three reviewers for providing thorough reviews of our work. We are confident that their feedback and suggestions have greatly improved the manuscript.

Since the changes involve new analyses and figures and subsequent rewriting of the text, we did not use the track changes function but used green text to highlight new text compared to the previous version of the manuscript. Deletions of text are shown as comments – see notes on these deletions at the end of this document.

Please see the point-by-point response to reviewers' comments.

Reviewer #1

Understanding forces shaping the evolution of genome architecture is a longstanding project for biology, and in their manuscript, Reinart et al provide compelling and novel insights into the evolution of repetitive sequences of neighboring genes. I was impressed by the scale and creativity of the analyses performed, which revealed a number of remarkable patterns across organisms. For example, the presence of repeats of neighboring genes appears to be a common feature, but the sequence composition of repeats found in gene boundaries can vary among taxa across the tree of life. The finding that these repeat features are linked closely to gene functions (i.e. essential vs environmentally responsive) was also revealing. The paper is well written and I think this will make a valuable publication, but I do have a few minor comments and one significant comment.

Thank you for the kind words regarding the work. We have made quite extensive changes to the work based on your suggestion to include phylogenetic data (see the major comment) and are confident this has improved the paper.

Line 15: 5,1 billion (is this 5.1?) elsewhere 5.1 billion is used, but “5.1” show up a few times.

You are correct in that “5,1” should be “5.1” at all places in the text. This is now fixed.

Figure 2b. I know it's in the legend, but labeling this as a “randomization” directly in the plot might be helpful for readers.

We have revised Figure 2 to show the randomly rearranged STR dataset as a histogram labeled as such.

Fig. 3. As nice looking as this is, it's hard to figure out where on the x-axis and y-axis the data line up with the 3-D view. I wonder if stacked panels for each taxa would be a clearer visualization. I think Extended Data Fig. 4/5 are close, but I imagine you could plot multiple lines on the same plot or similar so it can fit as a main text figure. (leave it to authors/editor, to decide if this is essential)

We agree that the 3-D figure can be difficult to read. In the revised Figure 4b, we show lines for two motifs per subpanel in 2-D.

Figure 5. Please label the subpanels with which nucleotide/dinucleotide is being plotted. It is tedious to scan back and forth from the legend to know what you are looking at in each panel.

Thank you for your suggestion. We have made labeling clearer in the revised Figure 5.

Some discussion from the authors on what mechanisms (DNA repair, natural selection, etc.) might be responsible for the enrichment of certain repeats in relation to genes across taxa would be interesting.

In the revised manuscript we speculate that the broad similarity of fungi/microeukaryotes and plants/animals in repeat profiles could be due to regulatory requirements due to genome size expansions

in plants/animals (L205-L206). Regarding natural selection, we note in the revised manuscript that the depletion for repetitiveness near housekeeping functions is likely due to negative selection (L240-L241).

Major: I was surprised the authors have not done any formal analyses or visualizations of the phylogenetic context of the patterns they are finding. I think adding something along these lines would enhance support for some of the conclusions. The authors should look for phylogenetic signal of various features (ie. Rx scores of specific nucleotides, etc) explicitly. For example see ape package in R, which have methods to test for phylogenetic signals, correlated evolution of traits, ancestral state reconstruction, etc. A time-calibrated phylogenetic tree for the species could be obtained from TimeTree 5. Formally testing for correlated evolution would give more credence to statements like Line 186: "These results show that the spatial patterns of repetitiveness have evolved in concert with mechanisms that drive the overall GC-content." While the results seem to suggest this, co-evolution of repetitiveness and GC-content can be formally tested in the context of a phylogeny. I think such analyses could be used in a number of places throughout this work and would provide statistical support for various comparisons across taxa. If this is new to the authors, I recommend "Analysis of Phylogenetics and Evolution with R"

Thank you for these suggestions. We had not considered using TimeTree as a source for the phylogenetic tree. In the revised manuscript we use a TimeTree phylogeny for:

- Revised Figure 1d: Enrichment/depletion of STRs within 100bp of gene annotations.
- Revised Figure 4a: Average repetitiveness in gene-proximal regions.
- Revised Figure 5: Shifts in repetitiveness considering annotated transcription sites.
- Revised Figure 7d: Depletion of housekeeping terms near high-scoring gene-proximal regions per species.
- Revised Supplementary Data Fig. 4-5: Comparison of median repetitiveness within and outside UTRs (and translation regions) with the median repetitiveness in the remaining gene-proximal region.

In addition, we used the TimeTree phylogeny to calculate Pagel's λ for average repetitiveness to formally assess the presence of a phylogenetic signal in relation to Revised Figure 4a (L86-L88; Supplementary Data Table 1).

We also used the TimeTree phylogeny to run phylogenetic generalized least-squares for phylogenetically informed correlations between genome-wide GC-content and variation in repetitiveness (L155-L157; Supplementary Fig. 9).

Note that in all these visualizations and tests, the sample size was reduced from 1,270 species to 891 species as we were unable to map all the species to TimeTree. This is noted in the text throughout the revised manuscript.

I tried to access the code the authors pointed to in their paper at

<https://figshare.com/s/0788da931cf2de7e3cc6>

Unfortunately, I received this when trying to view the code file:

{"message": "Insufficient permissions", "code": "InsufficientPermissions"}

We are sorry that the link we provided did not work. However, it seems that the other reviewers were able to access the Figshare repository using the provided link, so we encourage you to attempt again or maybe with a different web browser?

Reviewer #2

This manuscript reports a so-far missing analysis on short tandem repeats (STRs) around genes across eukaryotes. Authors have some exciting results: that STRs are more abundant than expected close to genes and match to transcription factor binding sites. Yet there are certain aspects that need to be

clarified so that the manuscript's value is good for publication.

Overall, I found the results of the TF analysis really interesting. If repetitiveness shifts around genes hold after showing 95% CI's, that's also great. I have multiple points regarding the GO Analysis. In general, it is not an easy read, because results often refer to extended figures and analyses are mostly not simple but then either they are not really backed up by stats or not clear how or why they are done the way they are done.

I did not find the manuscript ready for publication. I would love fellow repeat people publish their stuff in good journals. But there is some more work here. Hope this helps.

Thank you for reading and assessing the manuscript – your suggestions were very helpful. In response to your comments, we have included more statistics throughout the manuscript so that every pattern we discuss is backed up by stats. The GO-data has been reanalyzed based on your suggestion to use housekeeping genes directly (L168-L189), which hopefully provides more clarity.

Major Points.

1. Using the name STR:

STRs vary between 1-6 nucleotides in units. Sometimes monomers act differently than others. So just analyzing mono- and dimers is hardly a paper on STRs. I would suggest being more specific since it is misleading otherwise. Also, it wasn't clearly stated why authors did not include longer units of STRs.

We see your point regarding the naming of STRs and have written “monomeric and dimeric STRs” in the text where this is meant and provided reasoning and some statistics serving as rationale for our in-depth survey of monomer and dimer motifs: Revised Figure 1b now shows the number of all STRs (unit sizes 1-6) in the four major groups. Revised Figure 1c shows that monomers and dimers are very common in all groups and 1d shows that near gene borders (< 100 bp), monomer and dimer motifs tend to be enriched whereas less so for larger unit lengths. We hope these changes make it clear why we focus on monomeric and dimeric in the manuscript. See L69-L75.

2. Statistics

Author use some indirect way to measure presence of repeat units around genes. They call the measure repetitiveness. It is basically comparing the extent of repeating elements in 10 nucleotide long windows to the rest for each of upstream and downstream region. Unfortunately, the statistical tests are sometimes missing. For example, the biggest claim that the authors make (and I think it is true but we need to show it correctly) is that there are more repeats closer to transcription/translation start/end sites. But then there should be a statistics associated with this claim. And that is not clear. I list all such claims below:

Thank you for the input. We agree that the indirect way (i.e., controlling for the expected repetitiveness given base composition in gene-proximal regions) is perhaps not easy to grasp. The benefit of this metric is that it allowed us to produce T-statistics and P-values to assess the statistical significance of shifts in repetitiveness (as shown in revised Supplementary Fig. 3) and that it reduced the variance when comparing genomes with different base compositions with each other (evident by the narrow confidence interval bands in revised Figure. 4b). In the revised manuscript, the methods figure is moved to the main text. Concerning additional statistics, we show in the revised manuscript that conventional STR counting yield statistically significant enrichments near genes (revised Figure 1d) and provide a clearer visualization and statistics for the deviation between the expected spatial distribution of STR counts give random positioning and the observed spatial distribution (revised Fig. 2a-c).

- line 65: *Since you say the genomes varied, I would like to see some statistics test results for the Extended Data Figure 1. Maybe report it just below in the caption or as stars in the plot but a reporting of significance would be necessary.*

Thank you for this suggestion. We have included the F-statistics and P-values of ANOVA tests of equal means in the figure caption of Supplementary Fig. 1 (they were all highly significant).

- Extended Data Fig 4 and 5: All variation is plotted in 1 SD. It needs to be 2SD or 95% CI. I don't know if the patterns then hold. This I found worrisome, to be honest. Because if the 95% CI are too wide, then this would not be a helpful way to depict the varying abundance of repeats.

The 95% CIs of the data shown in Extended Data Fig 4-5 are now shown in the revised (main) Figure 4b (replacing the 3-D figure). Note that the 95% CIs are extremely narrow compared to the SD – and that the patterns hold.

- Extended Data Figure 6: *results are not corrected for multiple testing. Authors should then find another statistically sound way if correcting for too many tests here means too little power. There needs to be a correction. Otherwise no results can be deduced.*

We've now corrected for multiple testing using Bonferroni – all P-values remain significant except three assemblies (these assemblies are noted in Supplementary Fig. 3 caption).

-Fig 6: *'[if there are multiple assemblies per species, a random one is used.]' But for other analyses, average was taken. Why not here?*

We agree that there is no good reason for taking a random, single assembly versus using mean values from multiple assemblies in the models. Given problems with the analysis also noted by Reviewer 3, we reanalyzed the data (using a different statistical approach) taking mean values if multiple assemblies were available for one species.

- GO Analysis: *where are the statistical test results? Maybe I missed it. Based on the figure, I would not argue there are differences. Not obvious.*

The statistical test results were in the source data (.tsv file) of Figure 7 deposited at figshare. However, we see that this was not specifically noted in the caption or in the text. In the revised manuscript, the statistical test results can be found in Supplementary File 1 at <https://figshare.com/s/0788da931cf2de7e3cc6>.

- GO Analysis: line 494-495: *g profiler is used to test, all good but which statistical test was used, was a correction applied? these need to be described.*

A more thorough description of the statistical test and its correction can be found in the Methods section (L355-359).

- GO Analysis: Extended Data Fig 9: *this analysis is not helpful. I honestly did not understand, what we need to get out of it. And quantitatively, what it shows.*

Extended Data Fig. 9 concerned GC-content, so we assume here that you refer to Extended Data Fig. 10. We agree that the figure did not serve the intended purpose of visualizing genes with high STD in repetitiveness versus genes with low STD and we decided to remove it from the revised manuscript.

3. Clarity

Since the approaches authors take are not simple counts etc, I think clarity plays an important role. And there are certain things that are missing, which makes the text unclear. In conclusion for example, there are two points, where results are mentioned but no analysis are shown.

- Fig 2: Why has b higher counts than a? Because of the normalization? I think the amount of counts should be kept more even to convey the message.

Yes, it was due to the normalization performed separately for panel a and panel b. In revised Fig. 2 we show the data as histograms (a), including statistics (b) and with intergenic length information for real data only. For (c) we reworked the figure to ensure that the numbers were scaled according to the maximum STR counts per row (where each row is an intergenic length bin).

- Extended Data Fig 11: line 412-413: Why is this normalization used? What does it exactly do? Is it responsible of the effect I asked about above?

The intended normalization would avoid high counts skewed towards small intergenic regions as seen in Extended Data Fig. 11. As the intended purpose of the Extended Data Fig. 11 was unclear to two of the reviewers, we decided to remove it from the revised manuscript.

- Transcription Factor Binding Analysis: why is this analysis done only on seven species?

It was done on all species where we could access TRANSFAC data through g:Profiler, which was seven species. This is now noted in the revised text (L189-L191).

- Conclusion, lines 252-253: this was not shown. This would be a great analysis that I suggest authors do though. Compare the GO's for high variation genes with housekeeping genes. This would yield a significant result for sure and it would be an important discovery. I would prefer that result over what we see on Fig 7a since this would be more clear and statistically backed up.

Thank you for this great suggestion – we mapped enriched/depleted GO terms to GO terms of genes listed in Joshi et al. 2022 (a paper quantifying housekeeping genes) and tested for dependence between housekeeping function and depletion/enrichment of terms connected to high-STD repeat genes and led to a highly rewarding finding, see revised Figure 7c-d. Note that the housekeeping function terms are listed in Supplementary File 2.

- Conclusion: line 256-257: how exactly does it relate to Fig 7b? Is there a stretch regarding the chromatin structure or did I miss an analysis?

We did not perform chromatin structure analyses and agree that this formulation could be considered a stretch. We removed the phrasing “and thus shape the structure of chromatin” so the sentence reads “The repetitiveness of the STR motifs may dictate the arsenal of TFs that bind gene-proximal sequences to facilitate gene expression ... “ (L227-228).

- Methods: following questions would help with basic clarity:

How many STRs per genome are detected? how many are conserved?

We agree with this point and have revised Figure 1 to provide more clarity. Revised Fig. 1b shows the number of STRs per genome (per group) and revised Fig. 1c shows the top 40 STR motifs per group. Regarding conservation, we show in revised Fig. 1d that the enrichment of STRs within 100 bp of gene annotations is conserved for monomer and dimer STRs. We think that phylogenetic sorting (in multiple figures in the revised manuscript) makes it clear that the patterns of interest are conserved in the eukaryotic groups. Analysis of specific, conserved sites would require a different analytical framework and we consider it beyond the scope of this work.

What is GoaT exactly? Alignments?

GoaT is a database containing genome metadata (assembly id, assembly quality metrics, taxonomy, etc.) that we used to get metadata on each specific genome assembly we retrieved from Ensembl.

Why is ULTRA used but not others? TRF, Tral etc.

This was a simple matter of preference of the output (JSON output from ULTRA, and ULTRA outputs P-values). In our experience the repeats detected by ULTRA and TRF very much overlap.

Why not longer repeat units?

In the revised manuscript we have included some analysis on longer repeat units and added the results to the revised Figure 1b-d. Given the commonness of monomer and dimer motifs (revised Figure 1c) we find it natural to perform an in-depth survey of monomers and dimers (see L69-75) A second point is perhaps more practical, but the number of potential motifs grows large at higher unit sizes, and running detailed repetitiveness scans using all the potential motifs would exhaust our high-performance computing resources (<https://sigma2.no/nb>) for a dataset of this size.

Minors:

- line 75-76, move the (Extended Data Figure 2) up in the sentence maybe after (UTRs) since it describes the missing data to be more precise.

Thank you, we made this change in the text (L78).

- Fig 6: same color are used for species and variable, which becomes confusing. (maybe color only based on species?) The way how scaling is explained in the caption can be clearer.

In the revised manuscript, we redid this analysis (revised Figure 6a-b) and changed the color scheme.

- Extended Data Figure 7: Each line is what? Please mention it in the caption.

We added this information to the caption in revised Supplementary Figure 6.

Further suggestions:

- Fig 5: maybe use more contrasting colors. That would help to convey the message. And make the white dots bigger? Hard to see them, too.

We revised Figure 5 to show the TimeTree phylogeny, increased the size of the white dots, and chose a color scheme with better contrast.

- Abstract: For the first two refs, which describe emergence of STRs' role on gene expression in genomes, the following first genome' wide publications would be more accurate: ours: Bilgin Sonay and Carvalho, 2015 and our collaborators: Willems et al., 2016

Thank you for suggesting these references now included in the revised manuscript (L11).

- Extended Data Fig 3: very helpful figure. I would consider putting this figure in the main text since it is hard to follow how R is calculated.

We have considered this suggestion and now include a revised version of the Extended Data Fig. 3 in the main text (revised Fig. 3).

Reviewer #3:

Reinar et al report a study of mono- and di-nucleotide repeat variation across a large number of taxa.

Thank you for reading the manuscript and providing valuable comments – we hope that our responses here will lessen your reservedness regarding the repetitiveness metric.

Major:

1. The definition of STRs in the manuscript needs clarification. The study is limited to repeating mono and dinucleotide sequences as explained in the Methods, but there does not appear to be a statement as to what constitutes a minimum size of a mononucleotide STR. Reading through the manuscript it is not clear whether the sequences 'AA' or 'AAA' might be identified as a STR in this work. The wording of the manuscript should be updated throughout to clarify this, and some justification as to this choice would be useful - the STRs may not reflect TF binding site lengths, and most (if not all) disease-causing STR expansions are trinucleotide or greater, for example, and it would be useful to see discussion of this.

In the revised manuscript, we refer to sites detected by ULTRA as STRs and use motif repetitiveness otherwise. Indeed, the repetitiveness of motif A in a sequence containing AA would be larger than zero, but it would not be recorded or referred to as an STR.

Regarding TF-binding it is true that not all repetitive stretches we detected match the length of known TF binding sites but a lot of them do. In the revised manuscript we more directly show that gene-proximal repetitiveness correlates to repetitiveness in TF binding sites (revised Figure 7e).

Regarding disease, we note this point in the Discussion of the revised manuscript (L210-212).

2. The methodology in this version of the manuscript is unclear. The spatial analysis of whole genomes section appears to describe identification of the position of mononucleotide and dinucleotide sequences with ULTRA, then the positions are then randomly assigned to the same intergenic region in a 'mock STR' whose purpose is unclear - the word 'mock' is never mentioned again. The 'normalise_axis' parameter of Vaex is used, but the actual function is never described; what normalisation is performed? Are the normalised positions used elsewhere?

We generated a baseline expectation of STR positioning in all the genome assemblies given random positioning (but restricted to the chromosome/scaffold/contig) where it was detected, which we termed the 'mock STR' control. We have updated revised Figure 1 to provide more clarity on what the 'mock STR' control shows compared to real data. We also utilized the 'mock STR' dataset in additional analyses present in the revised manuscript (revised Figure 4d and revised Figure 2a-b) to show how extreme the presence of observed STRs is within 100 bp of annotated gene bodies compared to expectations given random positioning. See also the *Methods* sections “*Enrichment/depletion of STRs within 100 bp of gene regions*” and “*Deviations from expected STR counts*” in the revised manuscript.

Regarding the 'normalize_axis' parameter we omitted this from the revised manuscript and wrote the function for normalization manually, to ensure that the bright green coloring reflects the highest value detected along the possible positions in the intergenic regions.

3. The authors propose a metric of repetitiveness that requires significant elaboration and is not well-explained in the Python notebook. Consider the sequences 'ACACACACAC' and 'ACACATACAC', which differ by a C->T substitution at one position. It would appear that the latter would be considered to be twice as repetitive as the former by merit of containing two AC runs, despite the sequence differing by only one nucleotide. Examples like 'AAAAAAAAA' vs 'AATAATAAT' show similar behaviour, with the latter apparently being labelled three times as repetitive as the first.

We have sought to better explain the repetitiveness metric in the revised text and have revised Figure 3 as a main figure. Note that in your examples, AC would get a repetitiveness-score of 1.0 in the sequence ACACACACAC and ACACATACAC would get a score of 0.8 as 80% of its nucleotides are within a repetitive stretch. AAAAAAAAAA gets a score of 1.0 and AATAATAAT a score of 0.66. In the revised Jupyter Notebook, we show examples of AT-repetitiveness as well. We hope our revised description of the metric is helpful and improves the manuscript.

4. Fig 3. is difficult to read. It shows some changes between clades, but there is no indication of the significance of such changes and the use of 3D projections make it difficult to read the figure. Some comment as to what is happening here in the context of known biology would be useful but is absent - for example, what is happening with TATA box elements, and is there a corresponding AT signal? Would these be considered STRs?

We agree that 3-D is challenging and have replaced it with a 2-D figure (revised Figure 4b) showing 95% confidence intervals. The TATA box would not be considered an STR, but it would slightly increase the average repetitiveness of a window if spatially conserved in enough species. However, from revised Fig. 5 where the average location of transcription sites per genome is located it does not seem to be any overlap between the approximate -30 position upstream of the transcription start and a peak of AT repetitiveness. We discuss this briefly in the revised manuscript (L206-L209).

5. Fig 5. is extremely dense and would be clearer if reworked. The sites also appear to have been independently hierarchically clustered, which scrambles the orders across plots - as a result, it's not possible to see whether the changes in repetitiveness in mononucleotide A STRs are the inverse of those in mononucleotide T STRs.

We have sought to improve the clarity in revised Fig. 5 by capping the maximum/minimum values within a common range, changing the coloring scheme, increasing the size of the transcription site annotations, and showing phylogenetically sorted data.

6. Insufficient detail is provided for training of the gradient-boosted trees. The prediction targets should be properly defined in text, not solely in Fig 6, and any imbalance in target classes discussed along with strategies for addressing these. Hyperparameter selection should be explained. Calculation of robustness of predictor performance ideally would use a clearly described cross-validation strategy or a totally held-out subset of data rather than a reused fold of cross-validation.

The objective of this analysis was to quantify the patterns that are clearly present when looking at Figure 5. As we in the revised version now use phylogenetic trees to visualize the data, we realize that a simple test of differences between sequence repetitiveness upstream and within the annotated 5'UTR sequence (and between sequence repetitiveness downstream of the 3'UTR and within the annotated 3'UTR sequence) visualized along the phylogeny is actually sufficient, without requiring to estimate hyperparameters etc. The result of the new analyses is shown in revised Figure 6a-b and as we see it perhaps even better quantifies and reflects the patterns evident in Figure 5.

7. Figure 7a appears to show no signal or discriminatory capacity on the axes - word clouds are not a good choice for visualisation here, and the use of too high a level of the ontology means that all axes appear to have 'response', or 'regulation', the meaning of which is unclear in the context of this plot. 7b appears to show that 'AT' presence is correlated in the opposite direction to 'CG' presence for CG-rich TF binding sites, which would appear to be a truism, but this is not commented on.

Following a suggestion by Reviewer 2 we have reanalyzed GO terms (including depletion in addition to enrichment) and defined a set of housekeeping functions based on gene lists published by Joshi et al. 2022 to assess if there is a difference in housekeeping function between the top 10% genes ranked by variation in repetitiveness and background sets of genes (revised Fig. 7e). We are confident that these new analyses and the reorganization of Figure 7 to include a bit of methods provided clearer results.

8. I would ideally like to see a more-thorough review of the existing literature of STR spatial distribution and more in-depth contextualisation of the key results here in the context of that body of work. The manuscript would also benefit from a clear discussion of how known features (RNA Pol II binding site motifs, for example) show in the metrics the authors define.

Regarding the spatial distribution of STRs very little is known, which was the major rationale for this work. Known features such as RNA Pol II binding sites would only appear if they were sufficiently spatially conserved in relation to translation start sites, which does not seem to be the case, but this cannot be completely ruled out (revised: see L206-L209)).

9. Additional detail is needed for the enrichment analysis parameters and the specific background gene lists used should be listed, and reducing the dimensionality to two dimensions is potentially oversimplifying the data - an approach with less reduction that selects the PCs explaining the majority of variation may be more appropriate.

We've added details on the specific background gene lists (L357) and in the revised Fig. 7 we no longer apply PCA in the reanalysis of the GO results but test for a dependence between repetitiveness and housekeeping functions. We are confident that the reanalysis provided clearer results.

Minor:

1. Commas are used in place of decimal points in numbers throughout text.

Thank you for pointing this out. This is now fixed.

2. Extended Data Table 1 and Extended Data Table 2 do not contain data; they should probably be listed as a supplementary figure that shows the structure of the data, not as data itself.

You are right in this but due to revision of the GO analysis Extended Data Table 2 is no longer included in the work.

3. Extended Figure 11 is a solid blue box.

Due to our revision of Fig. 2 this figure is no longer needed or included in the revised manuscript.

4. Extended Figure 12 features illegible text at screen resolution.

Due to our revision of Fig. 7 this figure is no longer needed/included in the revised manuscript.

Reviewer #3 (Remarks on code availability):

I manually reviewed but did not install and run the code. I have reservations about the appropriateness of the repetitiveness metric and would like to see this explained in text. No README file is provided, but I expect that the Python notebook would execute from manual review.

We hope our response to major point 3, our descriptions of the metric (L80-L97), and additional examples with the repetitiveness function in the Jupyter Notebook will better illustrate the repetitiveness metric.

Notes on larger text deletions present in the revised manuscript:

L15: In the **Abstract**, “*and that statistical learning models point to these shifts as good predictors of transcription and translation site positions.*” was deleted from the text as the method we use to quantify this in the revised manuscript is not statistical learning.

L78: In the **Results**, “*To assert that the strong spatial pattern did not result from systematic artifacts in genome assemblies and gene annotations, the STR sites were randomly rearranged within their respective regions (contig, scaffold, or chromosome model) without altering their lengths or composition, and the spatial analysis was repeated. The negative control returned no spatial pattern (Fig. 2a-b).*” is essentially communicated in the Methods section “Spatial analysis of STRs in whole-genome assemblies” and was removed.

L216: In the **Discussion**, “was accurate as a predictor for the average location of transcription and translation sites in unseen data” was replaced with “Gene-proximal repetitiveness marked transcription and translation boundaries in a group-specific manner indicating that the repetitiveness around transcription and translation sites is conserved within taxa” to better reflect the nature of the applied statistics.

Point-by-point response to reviewers' comments

Reviewer #1 (Remarks to the Author):

The authors have fully addressed all of my comments. My only remaining note is to encourage the authors to make sure the figure font sizes are legible per journal requirements (minimum size 6, I believe?)

Thank you for reading the revised manuscript. We are happy that our responses adequately addressed your comments. Regarding font sizes in the figures, we've made sure they fall in the range 5-7 according to the journal guidelines.

I was also asked to comment on the authors response to Reviewer #2. While I think the reviewer has more expertise in satellite/repeat evolution in general than myself, I found the authors responses and revisions valuable. In particular, they have significantly improved the statistical rigor of their study, which was a primary concern from Reviewer #2. I find their responses acceptable and do not have further suggestions with respect to reviewer #2's comments.

Thank you for evaluating our responses to Reviewer #2.

Reviewer #1 (Remarks on code availability):

The authors provide a Jupyter notebook for running code and the datasets.

Reviewer #3 (Remarks to the Author):

Thank you for your revisions - nearly all of my points have been excellently addressed. Congratulations on a very nice paper! There is one minor change that I feel should be addressed further.

Regarding the response to point 8 of "Regarding the spatial distribution of STRs very little is known, which was the major rationale for this work.":

There is a moderate body of literature relating to the inter-species conservation and differences in length and spatial distribution of these elements that should be contextualised here. This appears in the literature as STRs, microsatellites, and sometimes k-mers.

Some examples that might be useful to include are:

<https://bmcgenomics.biomedcentral.com/articles/10.1186/s12864-019-5516-5>

<https://www.ncbi.nlm.nih.gov/pmc/articles/PMC5054066/#bib36>

<https://pubmed.ncbi.nlm.nih.gov/36289560/>

<https://www.mdpi.com/2073-4425/12/10/1571>

<https://genome.cshlp.org/content/13/10/2242>

<https://www.nature.com/articles/ng0795-337>

Thank you! We have revised the start of the Discussion to include three of the papers you list (note that the Srivastava paper was referenced in the introduction). We sought to address the heterogeneity of STR definitions (highlighted in the Verbieest et al. 2023 review and here we also included the Tørresen et al. 2019 review) and the known link between STR content and GC-content (Dieringer and Schlötterer 2003). The Sievers et al. 2021 paper was new to us, but it seems like their k-mer approach revealed patterns aligning with ours, as they reported that monomer and dimer STRs are responsible for conserved patterns both within kingdoms and between kingdoms, which is reflected by our results – and was accordingly included in the Discussion.